# Heterogeneity in transmissibility and shedding SARS-CoV-2 via droplets and aerosols

Paul Z Chen[1], Niklas Bobrovitz[2,3,4], Zahra Premji[5], Marion Koopmans[6], David N Fisman[7,8], Frank X Gu[1,9]*

[1]Department of Chemical Engineering & Applied Chemistry, University of Toronto, Toronto, Canada; [2]Temerty Faculty of Medicine, University of Toronto, Toronto, Canada; [3]Department of Critical Care Medicine, Cumming School of Medicine, University of Calgary, Calgary, Canada; [4]O'Brien Institute of Public Health, University of Calgary, Calgary, Canada; [5]Libraries & Cultural Resources, University of Calgary, Calgary, Canada; [6]Department of Viroscience, Erasmus University Medical Center, Rotterdam, Netherlands; [7]Division of Epidemiology, Dalla Lana School of Public Health, University of Toronto, Toronto, Canada; [8]Division of Infectious Diseases, Temerty Faculty of Medicine, University of Toronto, Toronto, Canada; [9]Institute of Biomedical Engineering, University of Toronto, Toronto, Canada

*For correspondence:
f.gu@utoronto.ca

Competing interests: The authors declare that no competing interests exist.

## Abstract

**Background:** Which virological factors mediate overdispersion in the transmissibility of emerging viruses remains a long-standing question in infectious disease epidemiology.

**Methods:** Here, we use systematic review to develop a comprehensive dataset of respiratory viral loads (rVLs) of SARS-CoV-2, SARS-CoV-1 and influenza A(H1N1)pdm09. We then comparatively meta-analyze the data and model individual infectiousness by shedding viable virus via respiratory droplets and aerosols.

**Results:** The analyses indicate heterogeneity in rVL as an intrinsic virological factor facilitating greater overdispersion for SARS-CoV-2 in the COVID-19 pandemic than A(H1N1)pdm09 in the 2009 influenza pandemic. For COVID-19, case heterogeneity remains broad throughout the infectious period, including for pediatric and asymptomatic infections. Hence, many COVID-19 cases inherently present minimal transmission risk, whereas highly infectious individuals shed tens to thousands of SARS-CoV-2 virions/min via droplets and aerosols while breathing, talking and singing. Coughing increases the contagiousness, especially in close contact, of symptomatic cases relative to asymptomatic ones. Infectiousness tends to be elevated between 1 and 5 days post-symptom onset.

**Conclusions:** Intrinsic case variation in rVL facilitates overdispersion in the transmissibility of emerging respiratory viruses. Our findings present considerations for disease control in the COVID-19 pandemic as well as future outbreaks of novel viruses.

**Funding:** Natural Sciences and Engineering Research Council of Canada (NSERC) Discovery Grant program, NSERC Senior Industrial Research Chair program and the Toronto COVID-19 Action Fund.

## Introduction

Severe acute respiratory syndrome coronavirus 2 (SARS-CoV-2) has spread globally, causing the coronavirus disease 2019 (COVID-19) pandemic with more than 129.2 million infections and 2.8

**eLife digest** To understand how viruses spread scientists look at two things. One is – on average – how many other people each infected person spreads the virus to. The other is how much variability there is in the number of people each person with the virus infects. Some viruses like the 2009 influenza H1N1, a new strain of influenza that caused a pandemic beginning in 2009, spread pretty uniformly, with many people with the virus infecting around two other people. Other viruses like SARS-CoV-2, the one that causes COVID-19, are more variable. About 10 to 20% of people with COVID-19 cause 80% of subsequent infections – which may lead to so-called superspreading events – while 60-75% of people with COVID-19 infect no one else.

Learning more about these differences can help public health officials create better ways to curb the spread of the virus. Chen et al. show that differences in the concentration of virus particles in the respiratory tract may help to explain why superspreaders play such a big role in transmitting SARS-CoV-2, but not the 2009 influenza H1N1 virus. Chen et al. reviewed and extracted data from studies that have collected how much virus is present in people infected with either SARS-CoV-2, a similar virus called SARS-CoV-1 that caused the SARS outbreak in 2003, or with 2009 influenza H1N1.

Chen et al. found that as the variability in the concentration of the virus in the airways increased, so did the variability in the number of people each person with the virus infects. Chen et al. further used mathematical models to estimate how many virus particles individuals with each infection would expel via droplets or aerosols, based on the differences in virus concentrations from their analyses. The models showed that most people with COVID-19 infect no one because they expel little – if any – infectious SARS-CoV-2 when they talk, breathe, sing or cough. Highly infectious individuals on the other hand have high concentrations of the virus in their airways, particularly the first few days after developing symptoms, and can expel tens to thousands of infectious virus particles per minute. By contrast, a greater proportion of people with 2009 influenza H1N1 were potentially infectious but tended to expel relatively little infectious virus when the talk, sing, breathe or cough.

These results help explain why superspreaders play such a key role in the ongoing pandemic. This information suggests that to stop this virus from spreading it is important to limit crowd sizes, shorten the duration of visits or gatherings, maintain social distancing, talk in low volumes around others, wear masks, and hold gatherings in well-ventilated settings. In addition, contact tracing can prioritize the contacts of people with high concentrations of virus in their airways.

million deaths (as of 1 April 2021) (*Dong et al., 2020*). For respiratory virus transmission, airway epithelial cells shed virions to the extracellular fluid before atomization (from breathing, talking, singing, coughing and aerosol-generating procedures) partitions them into a polydisperse mixture of particles that are expelled to the ambient environment. Aerosols ($\leq$100 µm) can be inhaled nasally, whereas droplets (>100 µm) tend to be excluded (*Prather et al., 2020*; *Roy and Milton, 2004*). For direct transmission, droplets must be sprayed ballistically onto susceptible tissue (*Liu et al., 2017a*). Hence, droplets predominantly deposit on nearby surfaces, potentiating indirect transmission. Aerosols can be further categorized based on typical travel characteristics: short-range aerosols (50–100 µm) tend to settle within 2 m; long-range ones (10–50 µm) often travel beyond 2 m based on emission force; and buoyant aerosols ($\leq$10 µm) remain suspended and travel based on airflow profiles for minutes to many hours (*Liu et al., 2017a*; *Wei and Li, 2015*). Although proximity has been associated with infection risk for COVID-19 (*Chu et al., 2020*), studies have also suggested that long-range airborne transmission occurs conditionally (*Hamner et al., 2020*; *Lu et al., 2020a*; *Park et al., 2020*).

While the basic reproductive number has been estimated to be 2.0–3.6 (*Hao et al., 2020*; *Li et al., 2020a*), transmissibility of SARS-CoV-2 is highly overdispersed (dispersion parameter $k$, 0.10–0.58), with numerous instances of superspreading (*Hamner et al., 2020*; *Lu et al., 2020a*; *Park et al., 2020*) and few cases (10–20%) causing many secondary infections (80%) (*Bi et al., 2020*; *Endo et al., 2020*; *Laxminarayan et al., 2020*). Similarly, few cases drive the transmission of SARS-CoV-1 ($k$, 0.16–0.17) (*Lloyd-Smith et al., 2005*), whereas influenza A(H1N1)pdm09 transmits more homogeneously ($k$, 7.4–14.4) (*Brugger and Althaus, 2020*; *Roberts and Nishiura, 2011*), despite

both viruses spreading by contact, droplets and aerosols (*Cowling et al., 2013*; *Yu et al., 2004*). Although understanding the determinants of viral overdispersion is crucial towards characterizing transmissibility and developing effective strategies to limit infection (*Lee et al., 2020*), mechanistic associations for *k* remain unclear. As an empirical estimate, *k* depends on myriad extrinsic (behavioral, environmental and invention) and host factors. Nonetheless, *k* remains similar across distinct outbreaks for a virus (*Lloyd-Smith et al., 2005*), suggesting that intrinsic virological factors mediate virus overdispersion.

Here, we investigated how intrinsic case variation in respiratory viral loads (rVLs) facilitates overdispersion in SARS-CoV-2 transmissibility. By systematic review, we developed a comprehensive dataset of rVLs from cases of COVID-19, SARS and A(H1N1)pdm09. Using comparative meta-analyses, we found that heterogeneity in rVL was associated with overdispersion among these emerging infections. To assess potential sources of case heterogeneity, we analyzed SARS-CoV-2 rVLs across age and symptomatology subgroups as well as disease course. To interpret the influence of heterogeneity in rVL on individual infectiousness, we modeled likelihoods of shedding viable virus via respiratory droplets and aerosols.

## Results

### Systematic review

We conducted a systematic review on virus quantitation in respiratory specimens taken during the infectious periods of SARS-CoV-2 ($-3$ to 10 days from symptom onset [DFSO]) (*Arons et al., 2020*; *He et al., 2020*; *Wölfel et al., 2020*), SARS-CoV-1 (0–20 DFSO) (*Pitzer et al., 2007*) and A(H1N1) pdm09 ($-2$ to 9 DFSO) (*Ip et al., 2017*) (Materials and methods). The systematic search (*Figure 1— source data 1*, *Figure 1—source data 2*, *Figure 1—source data 3*, *Figure 1—source data 4*, *Figure 1—source data 5*) identified 4274 results. After screening and full-text review, 64 studies met the inclusion criteria and contributed to the systematic dataset (*Figure 1*) ($N$ = 9631 total specimens), which included adult ($N$ = 5033) and pediatric ($N$ = 1608) cases from 15 countries and specimen measurements for asymptomatic ($N$ = 2387), presymptomatic ($N$ = 28) and symptomatic ($N$ = 7161) infections. According to a hybrid Joanna Briggs Institute critical appraisal checklist, risk of bias was low for most contributing studies (*Appendix 1—table 1*).

### Association of overdispersion with heterogeneity in rVL

We hypothesized that individual case variation in rVL facilitates the distinctions in *k* among COVID-19, SARS and A(H1N1)pdm09. For each study in the systematic dataset, we used specimen measurements (viral RNA concentration in a respiratory specimen) to estimate rVLs (viral RNA concentration in the respiratory tract) (Materials and methods). To investigate the relationship between *k* and heterogeneity in rVL, we performed a meta-regression using each contributing study (*Figure 2—figure supplement 1*), which showed a weak, negative association between the two variables (meta-regression slope *t*-test: p=0.038, Pearson's $r = -0.26$).

Using contributing studies with low risk of bias, meta-regression (*Figure 2*) showed a strong, negative association between *k* and heterogeneity in rVL for these three viruses (meta-regression slope *t*-test: p<0.001, Pearson's $r = -0.73$). In this case, each unit increase (one $\log_{10}$ copies/ml) in the standard deviation (SD) of rVL decreased log(*k*) by a factor of $-1.41$ (95% confidence interval [CI]: $-1.78$ to $-1.03$), suggesting that broader heterogeneity in rVL facilitates greater overdispersion in the transmissibility of SARS-CoV-2 than of A(H1N1)pmd09. To investigate mechanistic aspects of this association, we conducted a series of analyses on rVL and then modeled the influence of heterogeneity in rVL on individual infectiousness.

### Meta-analysis and subgroup analyses of rVL

We first compared rVLs among the emerging infections. We performed a random-effects meta-analysis (*Figure 2—figure supplement 2*), which approximated the expected rVL when encountering a COVID-19, SARS or A(H1N1)pdm09 case during the infectious period. This showed that the expected rVL of SARS-CoV-2 was comparable to that of SARS-CoV-1 (one-sided Welch's *t*-test: p=0.111) but lesser than that of A(H1N1)pdm09 (p=0.040).

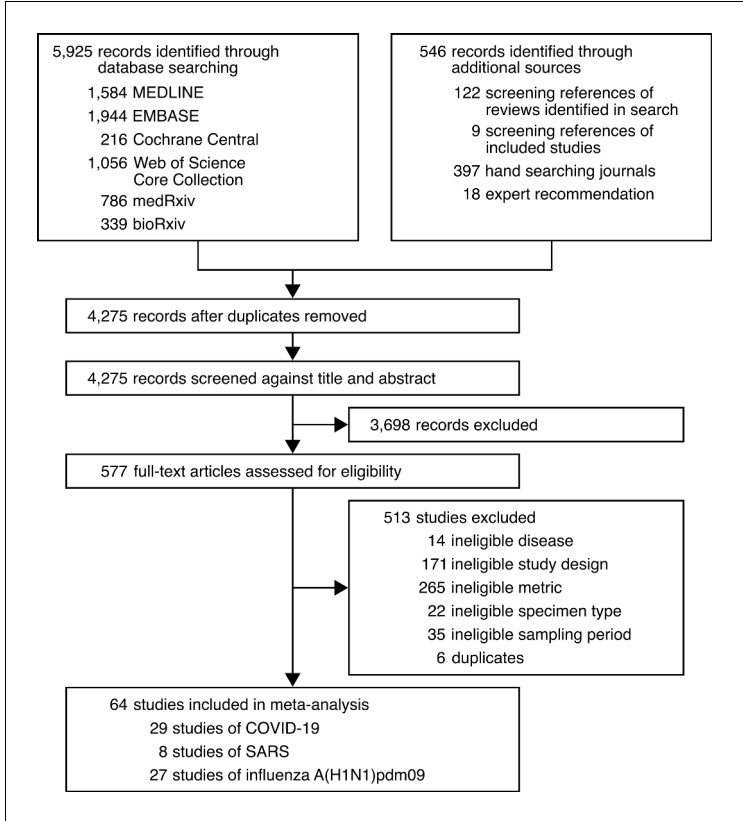

**Figure 1.** Development of the systematic dataset.
The online version of this article includes the following source data for figure 1:

**Source data 1.** Search strategy used for MEDLINE.
**Source data 2.** Search strategy used for EMBASE.
**Source data 3.** Search strategy used for Cochrane Central.
**Source data 4.** Search strategy used for Web of Science Core Collection.
**Source data 5.** Search strategy used for medRxiv and bioRxiv.

We also performed random-effects subgroup analyses for COVID-19 (**Figure 3**), which showed that expected SARS-CoV-2 rVLs were similar between pediatric and adult cases (p=0.476) and comparable between symptomatic/presymptomatic and asymptomatic infections (p=0.090). Since these meta-analyses had significant between-study heterogeneity among the mean estimates (Cochran's *Q* test: p<0.001 for each meta-analysis), we conducted risk-of-bias sensitivity analyses; meta-analyses of low-risk-of-bias studies continued to show significant heterogeneity (**Figure 3—figure supplements 1–5**).

## Distributions of rVL

We next analyzed rVL distributions. For all three viruses, rVLs best conformed to Weibull distributions (**Figure 4—figure supplement 1**), and we fitted the entirety of individual sample data for each virus in the systematic dataset (**Figure 4A**, **Figure 4—figure supplement 1N**). While COVID-19 and SARS cases tended to shed lesser virus than those with A(H1N1)pdm09 (**Figure 2—figure supplement 2**), broad heterogeneity in SARS-CoV-2 and SARS-CoV-1 rVLs inverted this relationship for highly infectious individuals (**Figure 4A**, **Figure 4—figure supplement 2A-C**). At the 90th case percentile (cp) throughout the infectious period, the estimated rVL was 8.91 (95% CI: 8.83–9.00) $\log_{10}$ copies/ml for SARS-CoV-2, whereas it was 8.62 (8.47–8.76) $\log_{10}$ copies/ml for A(H1N1)pdm09 (**Figure 4—figure supplement 3**). The SD of the overall rVL distribution for SARS-CoV-2 was 2.04 $\log_{10}$

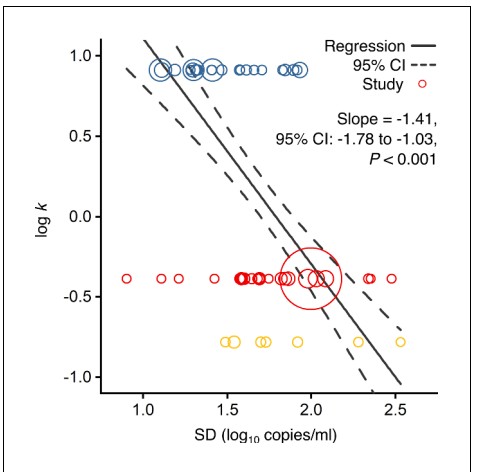

**Figure 2.** Association of overdispersion in SARS-CoV-2, SARS-CoV-1 and A(H1N1)pdm09 transmissibility with heterogeneity in respiratory viral load (rVL). Meta-regression of dispersion parameter ($k$) with the standard deviation (SD) of rVLs from contributing studies with low risk of bias (Pearson's $r = -0.73$). Pooled estimates of $k$ were determined from the literature for each infection. Blue, red and yellow circles denote A(H1N1)pdm09 ($N = 22$), COVID-19 ($N = 24$) and SARS ($N = 7$) studies, respectively. Circle sizes denote weighting in the meta-regression. The p-value was obtained using the meta-regression slope $t$-test. The online version of this article includes the following figure supplement(s) for figure 2:

**Figure supplement 1.** Meta-regression between dispersion in SARS-CoV-2, SARS-CoV-1 and A(H1N1) pdm09 transmissibility and heterogeneity in respiratory viral load (rVL).

**Figure supplement 2.** Meta-analysis of respiratory viral loads (rVLs) of SARS-CoV-2, SARS-CoV-1 and influenza A(H1N1)pdm09 during the estimated infectious period.

copies/ml, while it was 1.45 $\log_{10}$ copies/ml for A (H1N1)pdm09, showing that heterogeneity in rVL was indeed broader for SARS-CoV-2.

To assess potential sources of heterogeneity in SARS-CoV-2 rVL, we compared rVL distributions among COVID-19 subgroups. In addition to comparable mean estimates (*Figure 3*), adult, pediatric, symptomatic/presymptomatic and asymptomatic COVID-19 cases showed similar rVL distributions (*Figure 4*B, C), with SDs of 2.03, 2.06, 2.00 and 2.01 $\log_{10}$ copies/ml, respectively. Thus, age and symptomatology minimally influenced case variation in SARS-CoV-2 rVL during the infectious period.

## SARS-CoV-2 kinetics during respiratory infection

To analyze the influences of disease course, we delineated individual SARS-CoV-2 rVLs by DFSO and fitted the mean estimates to a mechanistic model for respiratory virus kinetics (*Figure 4D* and Materials and methods). The outputs indicated that, on average, each productively infected cell in the airway epithelium shed SARS-CoV-2 at 1.33 (95% CI: 0.74–1.93) copies/ml day$^{-1}$ and infected up to 9.25 susceptible cells (*Figure 4—figure supplement 4*). The turnover rate for infected epithelial cells was 0.71 (0.26–1.15) days$^{-1}$, while the half-life of SARS-CoV-2 RNA before clearance from the respiratory tract was 0.21 (0.11–2.75) days. By extrapolating the model to an initial rVL of 0 $\log_{10}$ copies/ml, the estimated incubation period was 5.38 days, which agrees with epidemiological findings (*Li et al., 2020a*). Conversely, the expected duration of shedding was 25.1 DFSO. Thus, SARS-CoV-2 rVL increased exponentially after infection, peaked around 1 DFSO along with the proportion of infected epithelial cells (*Figure 4—figure supplement 5*) and then diminished exponentially.

To evaluate case heterogeneity across the infectious period, we fitted distributions for each DFSO (*Figure 4E*), which showed that high SARS-CoV-2 rVLs also increased from the presymptomatic period, peaked at 1 DFSO and then decreased towards the end of the first week of illness. For the 90th cp at 1 DFSO, the rVL was 9.84 (95% CI: 9.17–10.56) $\log_{10}$ copies/ml, an order of magnitude greater than the overall 90th cp estimate. High rVLs between 1 and 5 DFSO were elevated above the expected values from the overall rVL distribution (*Figure 4—figure supplement 3*). At −1 DFSO, the 90th cp rVL was 8.30 (6.88–10.02) $\log_{10}$ copies/ml, while it was 7.93 (7.35–8.56) $\log_{10}$ copies/ml at 10 DFSO. Moreover, heterogeneity in rVL remained broad across the infectious period, with SDs of 1.83–2.44 $\log_{10}$ copies/ml between −1 to 10 DFSO (*Figure 4—figure supplement 2H-S*).

## Likelihood that droplets and aerosols contain virions

Towards analyzing the influence of heterogeneity in rVL on individual infectiousness, we first modeled the likelihood of respiratory particles containing viable SARS-CoV-2. Since rVL is an intensive quantity, the volume fraction of virions is low and viral partitioning coincides with atomization, we used Poisson statistics to model likelihood profiles. To calculate an unbiased estimator of

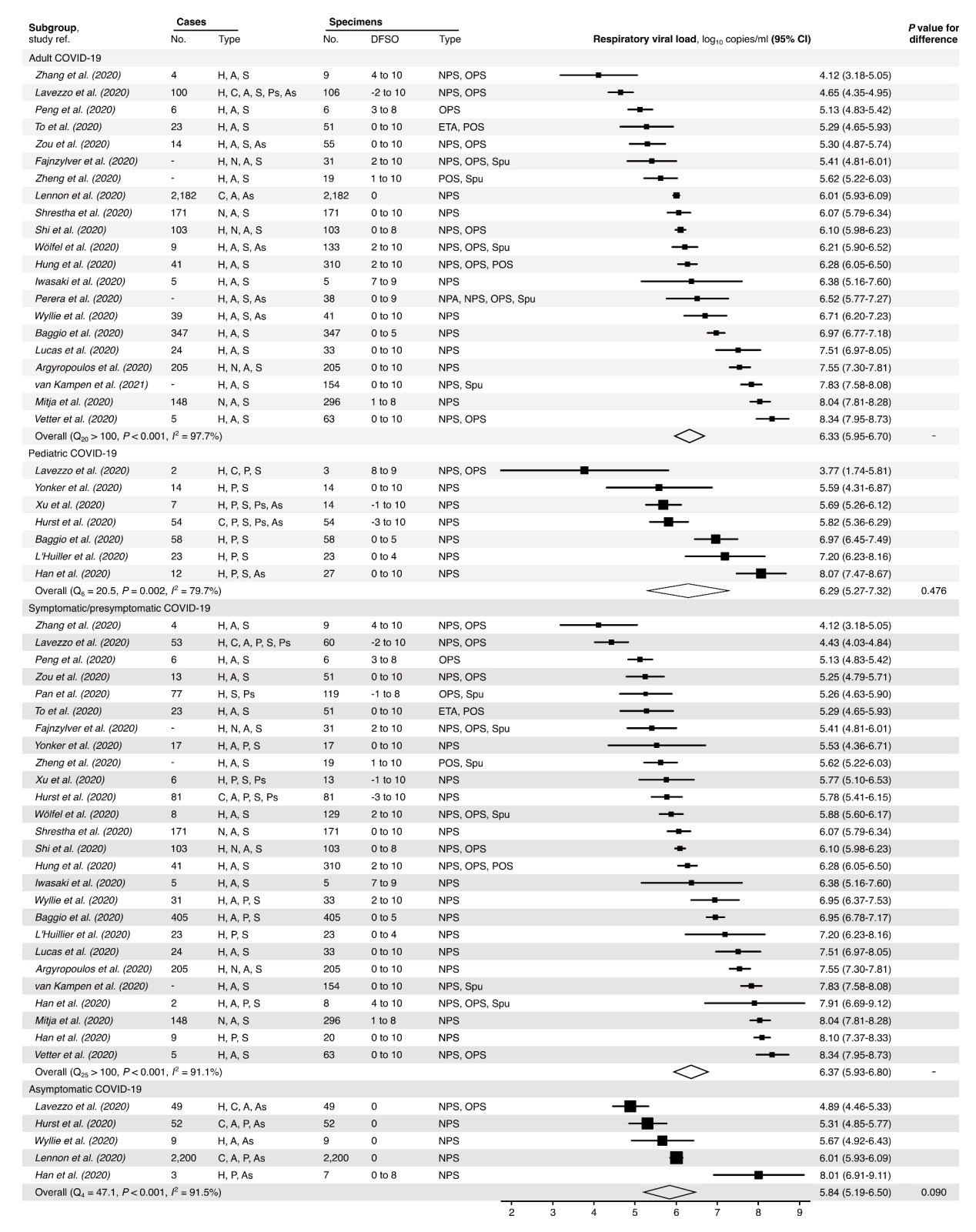

**Figure 3.** Subgroup analyses of SARS-CoV-2 respiratory viral load (rVL) during the infectious period. Random-effects meta-analyses comparing the expected rVLs of adult (≥18 years old) COVID-19 cases with pediatric (<18 years old) ones (top) and symptomatic/presymptomatic infections with asymptomatic ones (bottom) during the infectious period. Quantitative rVLs refer to virus concentrations in the respiratory tract. Case types: hospitalized (H), not admitted (N), community (C), adult (A), pediatric (P), symptomatic (S), presymptomatic (Ps) and asymptomatic (As). Specimen types:

*Figure 3 continued on next page*

*Figure 3 continued*

endotracheal aspirate (ETA), nasopharyngeal aspirate (NPA), nasopharyngeal swab (NPS), oropharyngeal swab (OPS), posterior oropharyngeal saliva (POS) and sputum (Spu). Dashes denote case numbers that were not obtained. Box sizes denote weighting in the overall estimates. Between-study heterogeneity was assessed using the p-value from Cochran's *Q* test and the $I^2$ statistic. One-sided Welch's *t*-tests compared expected rVLs between the COVID-19 subgroups (non-significance, p>0.05).

The online version of this article includes the following figure supplement(s) for figure 3:

**Figure supplement 1.** Risk-of-bias sensitivity analysis of between-study heterogeneity for SARS-CoV-2 respiratory viral load (rVL) during the estimated infectious period.

**Figure supplement 2.** Risk-of-bias sensitivity analysis of between-study heterogeneity for SARS-CoV-1 respiratory viral load (rVL) during the estimated infectious period.

**Figure supplement 3.** Risk-of-bias sensitivity analysis of between-study heterogeneity for A(H1N1)pdm09 respiratory viral load (rVL) during the estimated infectious period.

**Figure supplement 4.** Risk-of-bias sensitivity analysis of between-study heterogeneity for SARS-CoV-2 respiratory viral load (rVL) for adult COVID-19 cases during the estimated infectious period.

**Figure supplement 5.** Risk-of-bias sensitivity analysis of between-study heterogeneity for SARS-CoV-2 respiratory viral load (rVL) for symptomatic/presymptomatic COVID-19 cases during the estimated infectious period.

partitioning (the expected number of viable copies per particle), our method multiplied rVL estimates with particle volumes during atomization and an assumed viability proportion of 0.1% in equilibrated particles (Materials and methods).

When expelled by the mean COVID-19 case during the infectious period, respiratory particles showed low likelihoods of carrying viable SARS-CoV-2 (*Figure 5—figure supplement 1*). Aerosols (equilibrium aerodynamic diameter [$d_a$] ≤ 100 μm) were ≤3.16% (95% CI: 2.61–3.71%) likely to contain a virion. Droplets also had low likelihoods: at $d_a$ = 200 μm, they were 22.3% (21.4–23.2%), 3.36% (3.03–3.69%) and 0.34% (0.29–0.39%) likely to contain one, two or three virions, respectively.

COVID-19 cases with high rVLs, however, expelled particles with considerably greater likelihoods of carrying viable copies (*Figure 5A, B*, *Figure 5—figure supplement 1D, E*). For the 80th cp during the infectious period, aerosols ($d_a$ ≤ 100 μm) were ≤87.9% (95% CI: 87.2–88.5%) likely to carry at least one SARS-CoV-2 virion. For the 90th cp, larger aerosols tended to contain multiple virions (*Figure 5—figure supplement 1E*). At 1 DFSO, these estimates were greatest, and ≤98.8% (98.1–99.4%) of buoyant aerosols ($d_a$ ≤ 10 μm) contained at least one viable copy of SARS-CoV-2 for the 98th cp. When expelled by high cps, droplets ($d_a$ > 100 μm) tended to contain tens to thousands of SARS-CoV-2 virions (*Figure 5B, Figure 5—figure supplement 1E*).

## Shedding SARS-CoV-2 via respiratory droplets and aerosols

Using the partitioning estimates in conjunction with published profiles of the particles expelled by respiratory activities (*Figure 5—figure supplement 2*), we next modeled the rates at which talking, singing, breathing and coughing shed viable SARS-CoV-2 across $d_a$ (*Figure 5C-F*). Singing shed virions more rapidly than talking based on the increased emission of aerosols. Voice amplitude, however, had a significant effect on aerosol production, and talking loudly emitted aerosols at similar rates to singing (*Figure 5—figure supplement 2E*). Based on the generation of larger aerosols and droplets, talking and singing shed virions significantly more rapidly than breathing (*Figure 5C-E*). Each cough shed similar quantities of virions as in a minute of talking (*Figure 5C, F*).

Each of these respiratory activities expelled aerosols at greater rates than droplets, but particle size correlated with the likelihood of containing virions according to our model. Talking, singing and coughing expelled virions at comparable proportions via droplets (55.6–59.4%) and aerosols (40.6–44.4%), whereas breathing did so predominantly within aerosols (*Figure 5G*). Moreover, short-range aerosols mediated most of the virions (79.2–81.9%) shed via aerosols while talking normally and coughing. In comparison, while singing, or talking loudly, buoyant (14.5%) and long-range (17.5%) aerosols carried a larger proportion of the virions shed via aerosols (*Figure 5G*).

## Influence of heterogeneity in rVL on individual infectiousness

To interpret how heterogeneity in rVL influences individual infectiousness, we modeled total SARS-CoV-2 shedding rates (over all particle sizes) for each respiratory activity (*Figure 5H, Figure 5—figure supplement 3*). Between the 1st and the 99th cps, the estimates for a respiratory activity spanned ≥8.48 orders of magnitude on each DFSO; cumulatively from −1 to 10 DFSO, they

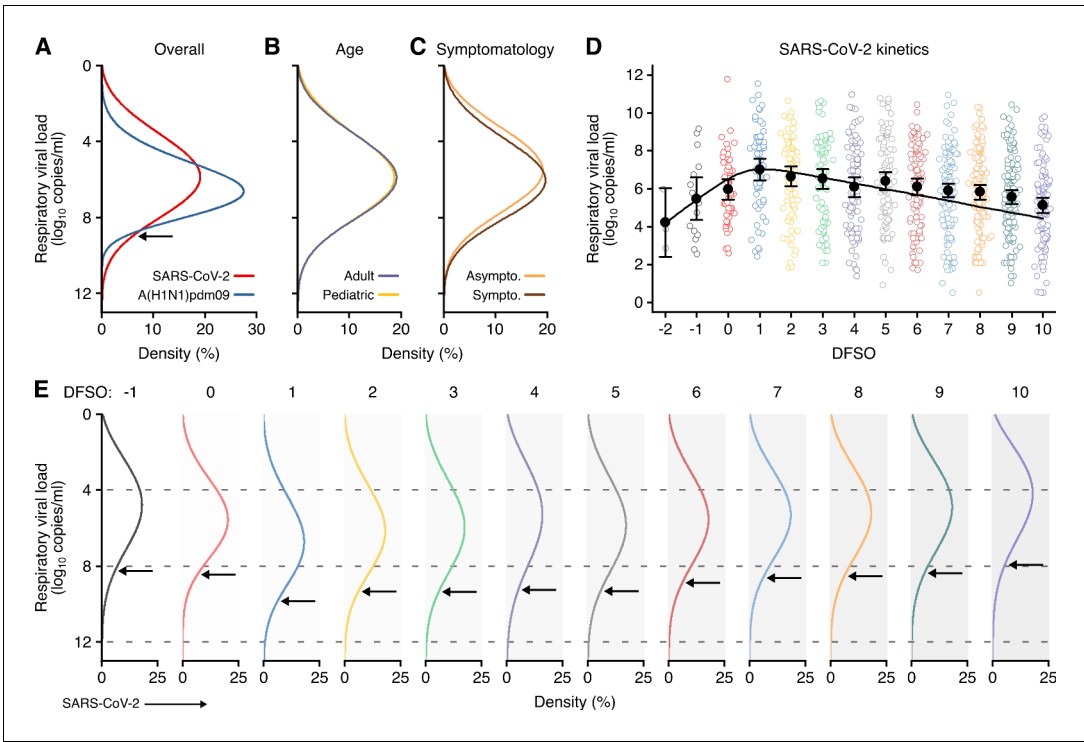

**Figure 4.** Heterogeneity and kinetics of SARS-CoV-2 respiratory viral load (rVL). (**A**) Estimated distribution of rVL for SARS-CoV-2 ($N$ = 3834 samples from $N$ = 26 studies) and A(H1N1)pdm09 ($N$ = 512 samples from $N$ = 10 studies) throughout the infectious periods. (**B, C**) Estimated distribution of SARS-CoV-2 rVL for adult ($N$ = 3575 samples from $N$ = 20 studies) and pediatric ($N$ = 198 samples from $N$ = 9 studies) (**B**) and symptomatic/ presymptomatic ($N$ = 1574 samples from $N$ = 22 studies) and asymptomatic ($N$ = 2221 samples from $N$ = 7 studies) (**C**) COVID-19 cases. (**D**) SARS-CoV-2 rVLs fitted to a mechanistic model of viral kinetics (black curve, $r^2$ = 0.84 for mean estimates). Filled circles and bars depict mean estimates and 95% confidence intervals. Open circles show the entirety of individual sample data over days from symptom onset (DFSO) (left to right, $N$ = 3, 15, 50, 63, 71, 75, 85, 93, 105, 136, 123, 128 and 115 samples from $N$ = 21 studies). (**E**) Estimated distributions of SARS-CoV-2 rVL across DFSO. Weibull distributions were fitted on the entirety of individual sample data for the virus, subgroup or DFSO in the systematic dataset. Arrows denote 90th case percentiles for SARS-CoV-2 rVL distributions.

The online version of this article includes the following figure supplement(s) for figure 4:

**Figure supplement 1.** Respiratory viral loads for SARS-CoV-2, SARS-CoV-1 and A(H1N1)pdm09 best conform to Weibull distributions.

**Figure supplement 2.** Case heterogeneity in respiratory viral loads (rVLs) across viruses, COVID-19 subgroups and disease course.

**Figure supplement 3.** Descriptive parameters for respiratory viral loads based on individual sample data.

**Figure supplement 4.** Model parameters describing SARS-CoV-2 kinetics during respiratory infection.

**Figure supplement 5.** Kinetics of SARS-CoV-2 and airway epithelial cells during respiratory infection.

spanned 11.0 orders of magnitude. Hence, many COVID-19 cases inherently presented minimal transmission risk, whereas highly infectious individuals shed considerable quantities of SARS-CoV-2. For the 98th cp at 1 DFSO, singing expelled 313 (95% CI: 37.5–3158) virions/min to the ambient environment, talking emitted 293 (35.1–2664) virions/min, breathing exhaled 1.54 (0.18–15.5) virions/min and coughing discharged 249 (29.8–25111) virions/cough; these estimates were approximately two orders of magnitude greater than those for the 85th cp. For the 98th cp at −1 DFSO, singing shed 14.5 (0.15–4515) virions/min and breathing exhaled $7.13 \times 10^{-2}$ ($7.20 \times 10^{-4}$–220.2) virions/min. The estimates at 9–10 DFSO were similar to these presymptomatic ones (*Figure 5H, Figure 5—figure supplement 3B*). As indicated by comparable mean rVLs (*Figure 3*) and heterogeneities in rVL (*Figure 4B, C*), adult, pediatric, symptomatic/presymptomatic and asymptomatic COVID-19 subgroups presented similar distributions for shedding virions through these activities.

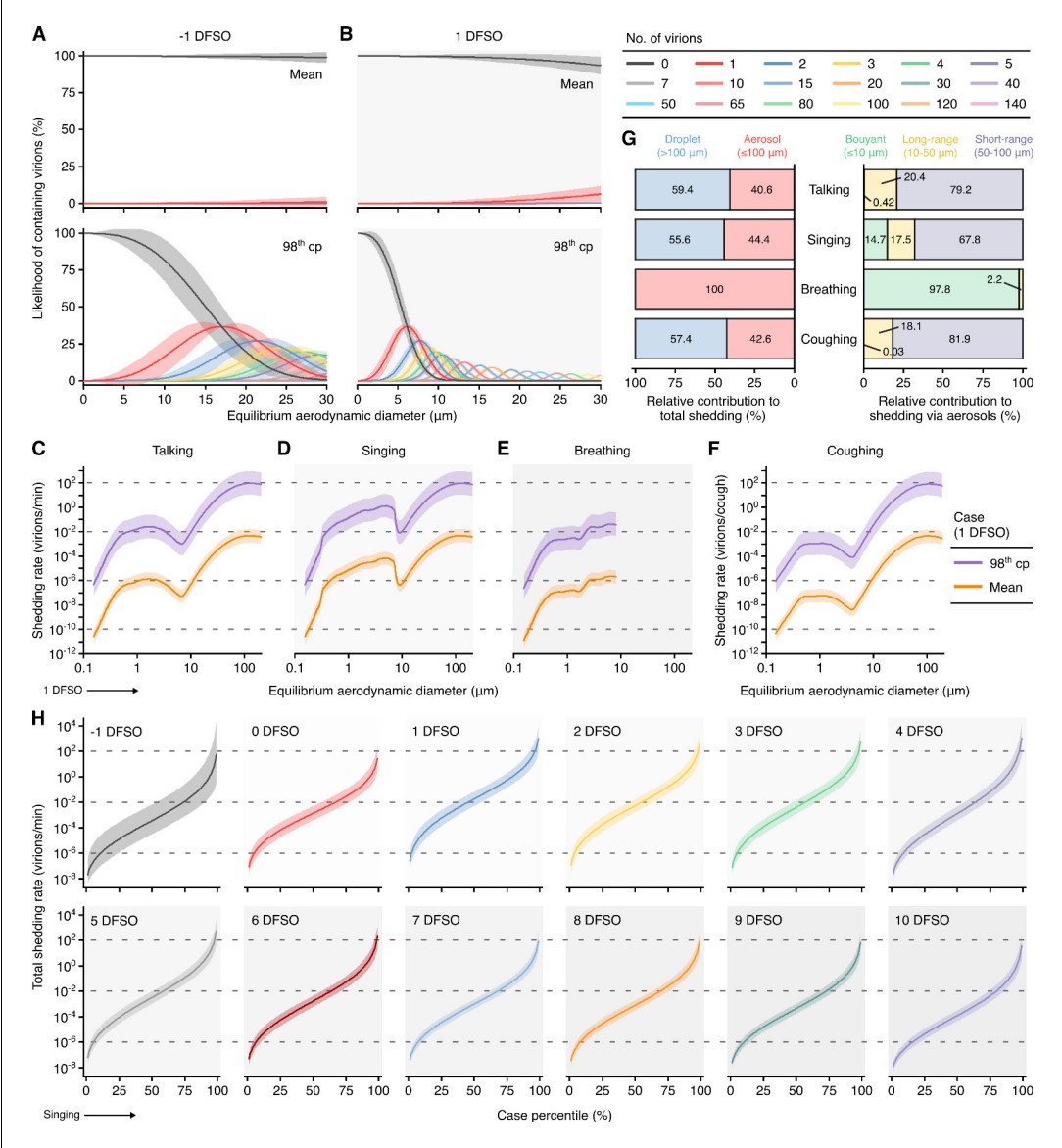

**Figure 5.** Heterogeneity in shedding SARS-CoV-2 via droplets and aerosols. (**A, B**) Estimated likelihood of respiratory particles containing viable SARS-CoV-2 when expelled by the mean (top) or 98th case percentile (cp) (bottom) COVID-19 cases at −1 (**A**) or 1 (**B**) days from symptom onset (DFSO). For higher number of virions, some likelihood curves were omitted to aid visualization. When the likelihood for zero virions approaches 0%, particles are expected to contain at least one viable copy. (**C–F**) Rate that the mean and 98th cp COVID-19 cases at 1 DFSO shed viable SARS-CoV-2 by talking, singing, breathing or coughing over particle size. (**G**) Relative contributions of droplets and aerosols to shedding virions for each respiratory activity (left). Relative contribution of buoyant, long-range and short-range aerosols to shedding virions via aerosols for each respiratory activity (right). (**H**) Case heterogeneity in the total shedding rate (over all particle sizes) of virions via singing across the infectious period. Earlier presymptomatic days were excluded based on limited data. Data range between the 1st and 99th cps. Lines and bands represent estimates and 95% confidence intervals, respectively, for estimated likelihoods or Poisson means.

The online version of this article includes the following figure supplement(s) for figure 5:

**Figure supplement 1.** Likelihood of respiratory particles containing SARS-CoV-2 or A(H1N1)pdm09.

**Figure supplement 2.** Rate profiles for particle expelled by respiratory activities.

**Figure supplement 3.** Heterogeneity in shedding SAR-CoV-2 via talking, breathing and coughing.

**Figure supplement 4.** Heterogeneity in infectiousness for COVID-19 and A(H1N1)pmd09 cases during the infectious period.

**Figure supplement 5.** Heterogeneity in shedding A(H1N1)pdm09 via droplets and aerosols.

We also compared the influence of case variation on individual infectiousness between A(H1N1)pdm09 and COVID-19. Aerosol spread accounted for approximately half of A(H1N1)pdm09 transmission events (*Cowling et al., 2013*), and the 50% human infectious dose for aerosolized influenza A virus is approximately 1–3 virions in the absence of neutralizing antibodies (*Fabian et al., 2008*). Based on the model, 62.9% of A(H1N1)pdm09 cases were infectious (shed $\geq$1 virion) via aerosols within 24 hr of talking loudly or singing (*Figure 5—figure supplement 4A*), and the estimate was 58.6% within 24 hr of talking normally and 22.3% within 24 hr of breathing. In comparison, 48.0% of COVID-19 cases shed $\geq$1 virion via aerosols in 24 hr of talking loudly or singing (*Figure 5—figure supplement 4C*). Notably, only 61.4% of COVID-19 cases shed $\geq$1 virion via either droplets or aerosols in 24 hr of talking loudly or singing (*Figure 5—figure supplement 4D*). While the human infectious dose of SARS-CoV-2 by any exposure route remains unelucidated, it must be at least one viable copy. Thus, at least 38.6% of COVID-19 cases were expected to present negligible risk to spread SARS-CoV-2 through either droplets or aerosols in 24 hr. The proportion of potentially infectious cases further decreased as the threshold increased: 55.8, 42.5 and 25.0% of COVID-19 cases were expected to shed $\geq$2, $\geq$10 and $\geq$100 virions, respectively, in 24 hr of talking loudly or singing during the infectious period.

While these analyses indicated that a greater proportion of A(H1N1)pdm09 cases were inherently infectious, 18.8% of COVID-19 cases shed virions more rapidly than those infected with A(H1N1)pdm09 (*Figure 4A*). At the 98th cp for A(H1N1)pdm09, singing expelled 4.38 (2.85–6.78) virions/min and breathing exhaled $2.15 \times 10^{-2}$ ($1.40 \times 10^{-2}$–$30.34 \times 10^{-2}$) virions/min. Highly infectious COVID-19 cases expelled virions at rates that were up to 1–2 orders of magnitude greater than their A(H1N1)pdm09 counterparts (*Figure 5H, Figure 5—figure supplement 5*).

## Discussion

This study provided systematic analyses of several factors characterizing SARS-CoV-2 transmissibility. First, our results indicate that broader heterogeneity in rVL facilitates greater overdispersion for SARS-CoV-2 than A(H1N1)pdm09. They suggest that many COVID-19 cases infect no one (*Bi et al., 2020*; *Endo et al., 2020*; *Laxminarayan et al., 2020*) because they inherently present minimal transmission risk via respiratory droplets or aerosols, although behavioral and environmental factors may further influence risk. Meanwhile, highly infectious cases can shed tens to thousands of SARS-CoV-2 virions/min, especially between 1 and 5 DFSO, potentiating superspreading events. The model estimates, when corrected to copies rather than virions, align with recent clinical findings for exhalation rates of SARS-CoV-2 (*Ma et al., 2020*). In comparison, a greater proportion of A(H1N1)pdm09 cases are infectious but shed virions at low rates, which concurs with more uniform transmission and few superspreading events observed during the 2009 H1N1 pandemic (*Brugger and Althaus, 2020*; *Roberts and Nishiura, 2011*). Moreover, our analyses suggest that heterogeneity in rVL may be generally associated with overdispersion for viral respiratory infections. In this case, rVL distribution can serve as an early correlate for transmission patterns, including superspreading, during outbreaks of novel respiratory viruses. When considered jointly with contact-tracing studies, this provides epidemiological triangulation on $k$: heterogeneity in rVL indirectly estimates $k$ via an association, whereas contact tracing empirically characterizes transmission chains to estimate $k$ but is limited by incomplete or incorrect recall of contact events by cases. When transmission is highly overdispersed, targeted interventions may disproportionately mitigate infection (*Lee et al., 2020*), with models showing that focused control efforts on the most infectious cases outperform random control policies (*Lloyd-Smith et al., 2005*).

Second, we analyzed SARS-CoV-2 kinetics during respiratory infection. While heterogeneity remains broad throughout the infectious period, rVL tends to peak at 1 DFSO and be elevated for 1–5 DFSO, coinciding with the period of highest attack rates observed among close contacts (*Cheng et al., 2020*). These results indicate that transmission risk tends to be greatest near, and soon after, illness rather than in the presymptomatic period, which concurs with large tracing studies (6.4–12.6% of secondary infections from presymptomatic transmission) (*Du et al., 2020*; *Wei et al., 2020*) rather than early temporal models (~44%) (*He et al., 2020*). Furthermore, our kinetic analysis suggests that, on average, SARS-CoV-2 reaches diagnostic concentrations 1.54–3.17 days after respiratory infection (−3.84 to −2.21 DFSO), assuming assay detection limits of 1–3 $\log_{10}$ copies/ml, respectively, for nasopharyngeal swabs immersed in 1 ml of transport media.

Third, we assessed the relative infectiousness of COVID-19 subgroups. As a common symptom of COVID-19 (*Guan et al., 2020*), coughing sheds considerable numbers of virions via droplets and short-range aerosols. Thus, symptomatic infections tend to be more contagious than asymptomatic ones, providing one reason as to why asymptomatic cases transmit SARS-CoV-2 at lower relative rates (*Li et al., 2020b*), especially in close contact (*Luo et al., 2020*), despite similar rVLs and increased contact patterns. Accordingly, children (48–54% of symptomatic cases present with cough) (*Lu et al., 2020b*; *Team and CDC COVID-19 Response Team, 2020*) may be less contagious than adults (68–80%) (*Guan et al., 2020*; *Team and CDC COVID-19 Response Team, 2020*) based on tendencies of symptomatology rather than rVL. Conversely, coughing sheds few virions via smaller aerosols. While singing and talking loudly, highly infectious cases can shed tens to hundreds of SARS-CoV-2 virions/min via long-range and buoyant aerosols.

Our study has limitations. The systematic search found a limited number of studies reporting quantitative specimen measurements from the presymptomatic period, meaning that these estimates may be sensitive to sampling bias. Although additional studies have reported semiquantitative metrics (cycle thresholds), these data were excluded because they cannot be compared on an absolute scale due to batch effects (*Han et al., 2021*), limiting use in compound analyses. In addition, our models considered virion partitioning during atomization to be a Poisson process, which stochastically associates partitioning with particle volume. Partitioning mechanisms associated with surface area, perhaps such as film bursting (*Bird et al., 2010*; *Johnson and Morawska, 2009*), may enrich the quantities of virions in smaller aerosols, based on their surface area-to-volume ratio. As severe COVID-19 is associated with high, persistent SARS-CoV-2 shedding in the lower respiratory tract (*Chen et al., 2021*) and small particles are typically generated there (*Johnson et al., 2011*), severe cases may also expel higher quantities of virions via smaller aerosols.

Furthermore, this study considered population-level estimates of the infectious periods, viability proportions and profiles for respiratory particles, which omit individual or environmental variation. Studies differ in their measurements of the emission rates and size distributions of the particles expelled during respiratory activities (*Johnson et al., 2011*; *Schijven et al., 2020*). Their characterization methods may prompt these differences, or they may be due to individual variation, including from distinctions in respiratory capacity, especially for young children, and phonetic tendencies (*Asadi et al., 2020*). Some patients shed SARS-CoV-2 with diminishing viability soon after symptom onset (*Wölfel et al., 2020*), whereas others produce replication-competent virus for weeks (*van Kampen et al., 2021*). The proportion of viable SARS-CoV-2 in respiratory particles, and how case characteristics or environmental factors influence it, remains under investigation (*Fears et al., 2020*; *Lednicky et al., 2020*; *Morris et al., 2020*). Cumulatively, these sources of variation may influence the shedding model estimates, further increasing heterogeneity in individual infectiousness.

Taken together, our findings provide a potential path forward for disease control. While talking, singing and coughing, our models indicate that SARS-CoV-2 is shed via droplets (55.6–59.4% of shed virions), short-range aerosols (30.1–34.9%), long-range aerosols (7.7–8.3%) and buoyant aerosols (0.01–6.5%). Transmission, however, requires exposure. For direct transmission, droplets tend to be sprayed ballistically onto susceptible tissue. Aerosols can be inhaled, may penetrate more deeply into the lungs and more easily facilitate superspreading events. However, with short durations of stay in well-ventilated areas, the exposure risk for both droplets and aerosols remains correlated with proximity to infectious cases (*Liu et al., 2017a*; *Prather et al., 2020*). Strategies to abate infection should limit crowd numbers and duration of stay while reinforcing distancing, low-voice amplitudes and widespread mask usage; well-ventilated settings can be recognized as lower-risk venues. Coughing can shed considerable quantities of virions, while rVL tends to peak at 1 DFSO and can be high throughout the infectious period. Thus, immediate, sustained self-isolation upon illness is crucial to curb transmission from symptomatic cases. Collectively, our analyses highlight the role of cases with high rVLs in propelling the COVID-19 pandemic. While diagnosing COVID-19, qRT-PCR can also triage contact tracing, prioritizing these patients: for nasopharyngeal swabs immersed in 1 ml of transport media, $\geq 7.14$ (95% CI: 7.07–7.22) $\log_{10}$ copies/ml corresponds to the top 20% of COVID-19 cases for variants before August 2020. Doing so may identify asymptomatic and presymptomatic infections more efficiently, a key step towards mitigation and elimination as the pandemic continues.

## Materials and methods

### Search strategy, selection criteria and data collection

We undertook a systematic review and prospectively submitted the protocol for registration on PROSPERO (registration number, CRD42020204637). Other than the title of this study, we have followed PRISMA reporting guidelines (*Moher et al., 2009*). The systematic review was conducted according to Cochrane methods guidance (*Higgins et al., 2019*).

The search included papers that (i) reported positive, quantitative measurements (copies/ml or an equivalent metric) of SARS-CoV-2, SARS-CoV-1 or A(H1N1)pdm09 in human respiratory specimens (endotracheal aspirate [ETA], nasopharyngeal aspirate [NPA], nasopharyngeal swab [NPS], oropharyngeal swab [OPS], posterior oropharyngeal saliva [POS] and sputum [Spu]) from COVID-19, SARS or A(H1N1)pdm09 cases; (ii) reported data that could be extracted from the estimated infectious periods of SARS-CoV-2 (defined as −3 to +10 DFSO for symptomatic cases and 0 to +10 days from the day of laboratory diagnosis for asymptomatic cases), SARS-CoV-1 (defined as 0 to +20 DFSO or the equivalent asymptomatic period) or A(H1N1)pdm09 (defined as −2 to +9 DFSO for symptomatic cases and 0 days to +9 days from the day of laboratory diagnosis for asymptomatic cases); and (iii) reported data for two or more cases with laboratory-confirmed COVID-19, SARS or A(H1N1)pdm09 based on World Health Organization (WHO) case definitions. Quantitative specimen measurements were considered after RNA extraction for diagnostic sequences of SARS-CoV-2 (*Ofr1b*, *N*, *RdRp* and *E* genes), SARS-CoV-1 (*Ofr1b*, *N* and *RdRp* genes) and A(H1N1)pdm09 (*HA* and *M* genes).

Studies were excluded, in the following order, if they (i) studied an ineligible disease; (ii) had an ineligible study design, including those that were reviews of evidence (e.g., scoping, systematic or narrative), did not include primary clinical human data, reported data for less than two cases due to an increased risk of selection bias, were incomplete (e.g., ongoing clinical trials), did not report an RNA extraction step before measurement or were studies of environmental samples; (iii) reported an ineligible metric for specimen concentration (e.g., qualitative RT-PCR or cycle threshold [Ct] values without calibration included in the study); (iv) reported quantitative measurements from an ineligible specimen type (e.g., blood specimens, pooled specimens or self-collected POS or Spu patient specimens in the absence of a healthcare professional); (v) reported an ineligible sampling period (consisted entirely of data that could not be extracted from within the infectious period); or (vi) were duplicates of an included study (e.g., preprinted version of a published paper or duplicates not identified by Covidence). We included data from control groups receiving standard of care in interventional studies but excluded data from the intervention group. Patients in the intervention group are, by definition, systematically different from general case populations because they receive therapies not being widely used for treatment, which may influence virus concentrations. Interventional studies examining the comparative effectiveness of two or more treatments were excluded for the same reason. Studies exclusively reporting semiquantitative measurements (e.g., Ct values) of specimen concentration were excluded as these measurements are sensitive to batch and instrument variation and, without proper calibration, cannot be compared on an absolute scale across studies (*Han et al., 2021*).

We searched, without the use of filters or language restrictions, the following sources: MEDLINE (via Ovid, 1946 to 7 August 2020), EMBASE (via Ovid, 1974 to 7 August 2020), Cochrane Central Register of Controlled Trials (via Ovid, 1991 to 7 August 2020), Web of Science Core Collection (including Science Citation Index Expanded, 1900 to 7 August 2020; Social Sciences Citation Index, 1900 to 7 August 2020; Arts & Humanities Citation Index, 1975 to 7 August 2020; Conference Proceedings Citation Index – Science, 1990 to 7 August 2020; Conference Proceedings Citation Index – Social Sciences & Humanities, 1990 to 7 August 2020; and Emerging Sources Citation Index, 2015 to 7 August 2020), as well as medRxiv and bioRxiv (both searched through Google Scholar via the Publish or Perish program, to 7 August 2020). We also gathered studies by searching through the reference lists of review articles identified by the database search, by searching through the reference lists of included articles, through expert recommendation (by Eric J. Topol and Akiko Iwasaki on Twitter) and by hand-searching through journals (*Nature*, *Nat. Med.*, *Science*, *NEJM*, *Lancet*, *Lancet Infect. Dis.*, *JAMA*, *JAMA Intern. Med.* and *BMJ*). A comprehensive search was developed by a librarian, which included subject headings and keywords. The search strategy had three main concepts (disease, specimen type and outcome), and each concept was combined using the appropriate Boolean operators. The search was tested against a sample set of known articles that were pre-

identified. The line-by-line search strategies for all databases are included in *Figure 1—source data 1*, *Figure 1—source data 2*, *Figure 1—source data 3*, *Figure 1—source data 4*, *Figure 1—source data 5*. The search results were exported from each database and uploaded to the Covidence online system for deduplication and screening.

Two authors independently screened titles and abstracts, reviewed full texts, collected data and assessed risk of bias via Covidence and a hybrid critical appraisal checklist based on the Joanna Briggs Institute (JBI) tools for case series, analytical cross-sectional studies and prevalence studies (*Moola et al., 2020*; *Munn et al., 2019*; *Munn et al., 2015*). To evaluate the sample size in a study, we used the following calculation:

$$n^* = \frac{z^2 \sigma}{d^2}, \tag{1}$$

where $n^*$ is the sample size threshold, $z$ is the z-score for the level of confidence (95%), $\sigma$ is the standard deviation (assumed to be 3 $\log_{10}$ copies/ml, one quarter of the full range of rVLs) and $d$ is the marginal error (assumed to be 1 $\log_{10}$ copies/ml, based on the minimum detection limit for qRT-PCR across studies) (*Johnston et al., 2019*). The hybrid JBI critical appraisal checklist is shown in the Appendix. Studies were considered to have low risk of bias if they met the majority of the items, indicating that the estimates were likely to be correct for the target population. Inconsistencies were resolved by discussion and consensus.

The search found 29 studies for COVID-19 (*Argyropoulos et al., 2020*; *Baggio et al., 2020*; *Fajnzylber et al., 2020*; *Han et al., 2020a*; *Han et al., 2020b*; *Hung et al., 2020*; *Hurst et al., 2020*; *Iwasaki et al., 2020*; *Kawasuji et al., 2020*; *L'Huillier et al., 2020*; *Lavezzo et al., 2020*; *Lennon et al., 2020*; *Lucas et al., 2020*; *Mitjà et al., 2020*; *Pan et al., 2020*; *Peng et al., 2020*; *Perera et al., 2020*; *Shi et al., 2020*; *Shrestha et al., 2020*; *To et al., 2020*; *van Kampen et al., 2021*; *Vetter et al., 2020*; *Wölfel et al., 2020*; *Wyllie et al., 2020*; *Xu et al., 2020*; *Yonker et al., 2020*; *Zhang et al., 2020a*; *Zheng et al., 2020*; *Zou et al., 2020*), 8 studies for SARS (*Chen et al., 2006*; *Cheng et al., 2004*; *Chu et al., 2005*; *Chu et al., 2004*; *Hung et al., 2004*; *Peiris et al., 2003*; *Poon et al., 2004*; *Poon et al., 2003*) and 27 studies for A(H1N1)pdm09 (*Chan et al., 2011*; *Cheng et al., 2010*; *Cowling et al., 2010*; *Duchamp et al., 2010*; *Esposito et al., 2011*; *Hung et al., 2010*; *Ip et al., 2016*; *Ito et al., 2012*; *Killingley et al., 2010*; *Launes et al., 2012*; *Lee et al., 2011a*; *Lee et al., 2011b*; *Li et al., 2010a*; *Li et al., 2010b*; *Loeb et al., 2012*; *Lu et al., 2012*; *Meschi et al., 2011*; *Ngaosuwankul et al., 2010*; *Rath et al., 2012*; *Rodrigues Guimarães Alves et al., 2020*; *Suess et al., 2010*; *Thai et al., 2014*; *To et al., 2010a*; *To et al., 2010b*; *Watanabe et al., 2011*; *Wu et al., 2012*; *Yang et al., 2011*), and data were collected from each study. For preprinted studies that were published as journal articles before the revised submission of this manuscript, we included the citation for the journal article. Descriptive statistics on quantitative specimen measurements were collected from confirmed cases directly if reported numerically or using WebPlotDigitizer 4.3 (https://apps.automeris.io/wpd/) if reported graphically. Individual specimen measurements were collected directly if reported numerically or, when the data were clearly represented, using the tool if reported graphically. We also collected the relevant numbers of cases, types of cases, reported treatments, volumes of transport media, numbers of specimens and DFSO (for symptomatic cases) or day relative to initial laboratory diagnosis (for asymptomatic cases) on which each specimen was taken. Hospitalized cases were defined as those being tested in a hospital setting and then admitted. Non-admitted cases were defined as those being tested in a hospital setting but not admitted. Community cases were defined as those being tested in a community setting. Symptomatic, presymptomatic and asymptomatic infections were defined as in the study. Based on rare description in contributing studies, paucisymptomatic infections, when described, were included with symptomatic ones. Pediatric cases were defined as those below 18 years of age or as defined in the study. Adult cases were defined as those 18 years of age or higher or as defined in the study.

## Calculation of rVLs from specimen measurements

In this study, viral concentrations in respiratory specimens were denoted as specimen measurements, whereas viral concentrations in the respiratory tract were denoted as rVLs. To determine rVLs, each collected quantitative specimen measurement was converted to rVL based on the dilution factor. For example, measurements from swabbed specimens (NPS and OPS) typically report the RNA

concentration in viral transport media. Based on the expected uptake volume for swabs ($0.128 \pm 0.031$ ml, mean ± SD) (*Warnke et al., 2014*) or reported collection volume for expulsed fluid in the study (e.g., 0.5–1 ml) along with the reported volume of transport media in the study (e.g., 1 ml), we calculated the dilution factor for each respiratory specimen to estimate the rVL. If the diluent volume was not reported, then the dilution factor was calculated assuming a volume of 1 ml (NPS and OPS), 2 ml (POS and ETA) or 3 ml (NPA) of transport media (*Lavezzo et al., 2020*; *Poon et al., 2004*; *To et al., 2020*). Unless dilution was reported for Spu specimens, we used the specimen measurement as the rVL (*Wölfel et al., 2020*). The non-reporting of diluent volume was noted as an element increasing risk of bias in the hybrid JBI critical appraisal checklist. Specimen measurements (based on instrumentation, calibration, procedures and reagents) are not standardized and, as DFSO is typically based on patient recall, there is also inherent uncertainty in these values. While the above procedures (including only quantitative measurements after extraction as an inclusion criterion, considering assay detection limits and correcting for specimen dilution) have considered many of these factors, non-standardization remains an inherent limitation in the variability of specimen measurements.

## Meta-regression of *k* and heterogeneity in rVL

To assess the relationship between *k* and heterogeneity in rVL, we performed a univariate meta-regression ($\log k = a{*}SD + b$, where *a* is the slope for association and *b* is the intercept) between pooled estimates of *k* (based on studies describing community transmission) for COVID-19 (*k* = 0.409) (*Adam et al., 2020*; *Tariq et al., 2020*; *Zhang et al., 2020b*; *Laxminarayan et al., 2020*; *Bi et al., 2020*; *Endo et al., 2020*; *Riou and Althaus, 2020*), SARS (*k* = 0.165) (*Lloyd-Smith et al., 2005*) and A(H1N1)pdm09 (*k* = 8.155) (*Brugger and Althaus, 2020*; *Roberts and Nishiura, 2011*) and the SD of the rVLs in contributing studies. Since SD was the metric, we used a fixed-effects model. For weighting in the meta-regression, we used the proportion of rVL samples from each study relative to the entire systematic dataset ($W_i = n_i/n_{total}$). All calculations were performed in units of $\log_{10}$ copies/ml. As the meta-regression used pooled estimates of *k* for each infection, it assumed that there was no correlated bias to *k* across contributing studies. The limit of detection for qRT-PCR instruments used in the included studies did not significantly affect the analysis of heterogeneity in rVL as these limits tended to be below the values found for specimens with low virus concentrations. The meta-regression was conducted using all contributing studies and showed a weak association. Meta-regression was also conducted using studies that had low risk of bias according to the hybrid JBI critical appraisal checklist and showed a strong association. The p-value for association was obtained using the meta-regression slope *t*-test for *a*, the effect estimate. While there is intrinsic measurement error in virus quantitation, based on the systematic review protocol and study design (as described above), this error should similarly increase heterogeneity in rVL for each virus, and the difference in heterogeneity in rVL between viruses should arise from the viruses.

## Meta-analysis of rVLs

Based on the search design and composition of contributing studies, the meta-analysis overall estimates were the expected SARS-CoV-2, SARS-CoV-1 and A(H1N1)pdm09 rVL when encountering a COVID-19, SARS or A(H1N1)pdm09 case, respectively, during their infectious period. Pooled estimates and 95% CIs for the expected rVL of each virus across their infectious period were calculated using a random-effects meta-analysis (DerSimonian and Laird method). For studies reporting summary statistics in medians and interquartile or total ranges, we derived estimates of the mean and variance and calculated the 95% CIs (*Wan et al., 2014*). All calculations were performed in units of $\log_{10}$ copies/ml. Between-study heterogeneity in meta-analysis was assessed using Cochran's *Q* test and the $I^2$ and $\tau^2$ statistics. If significant between-study heterogeneity in meta-analysis was encountered, sensitivity analysis based on the risk of bias of contributing studies was performed. The meta-analyses were conducted using STATA 14.2 (StataCorp LLC, College Station, TX, USA).

## Age and symptomatology subgroup analyses of SARS-CoV-2 rVLs

The overall estimate for each subgroup was the expected rVL when encountering a case of that subgroup during the infectious period. Studies reporting data exclusively from a subgroup of interest were directly included in the analysis after rVL estimations. For studies in which data for these

subgroups constituted only part of its dataset, rVLs from the subgroup were extracted to calculate the mean, variance and 95% CIs. Random-effects meta-analysis was performed as described above. For meta-analyses of pediatric and asymptomatic COVID-19 cases, contributing studies had low risk of bias, and no risk-of-bias sensitivity analyses were performed for these subgroups.

## Distributions of rVL

We pooled the entirety of individual sample data in the systematic dataset by disease, COVID-19 subgroups and DFSO. For analyses of SARS-CoV-2 dynamics across disease course, we included estimated rVLs from negative qRT-PCR measurements of respiratory specimens for cases that had previously been quantitatively confirmed to have COVID-19. These rVLs were estimated based on the reported assay detection limit in the respective study. Probability plots and modified Kolmogorov–Smirnov tests used the Blom scoring method and were used to determine the suitability of normal, lognormal, gamma and Weibull distributions to describe the distribution of rVLs for SARS-CoV-2, SARS-CoV-1 and A(H1N1)pdm09. For each virus, the data best conformed to Weibull distributions, which is described by the probability density function

$$f(v) = \frac{\alpha}{\beta}\left(\frac{v}{\beta}\right)^{\alpha-1} e^{-(v/\beta)^{\alpha}}, \tag{2}$$

where $\alpha$ is the shape factor, $\beta$ is the scale factor and $v$ is rVL ($v \geq 0$ $\log_{10}$ copies/ml). Weibull distributions were fitted on the entirety of collected individual sample data for the respective category. Since individual specimen measurements could not be collected from all studies, there was a small bias on the mean estimate for each fitted distribution. Thus, for the curves shown in *Figure 4B, C*, the mean of the Weibull distributions summarized in *Figure 4—figure supplement 2* was adjusted to be the subgroup meta-analysis estimate for correction; the SD and distribution around that mean remained consistent.

For each Weibull distribution, the value of the rVL at the $x$ th percentile was determined using the quantile function,

$$v_x = \beta[-\ln(1-x)]^{1/\alpha}. \tag{3}$$

For cp curves, we used *Equation (3)* to determine rVLs from the 1st cp to the 99th cp (step size, 1%). Curve fitting to *Equation (2)* and calculation of *Equation (3)* and its 95% CI was performed using the Distribution Fitter application in Matlab R2019b (MathWorks, Inc, Natick, MA, USA).

## Viral kinetics

To model SARS-CoV-2 kinetics during respiratory infection, we used a mechanistic epithelial cell-limited model for the respiratory tract (*Baccam et al., 2006*), based on the system of differential equations:

$$\frac{dT}{dt} = -\beta TV \tag{4}$$

$$\frac{dI}{dt} = \beta TV - \delta I \tag{5}$$

$$\frac{dV}{dt} = pI - cV, \tag{6}$$

where $T$ is the number of uninfected target cells, $I$ is the number of productively infected cells, $V$ is the rVL, $\beta$ is the infection rate constant, $p$ is the rate at which airway epithelial cells shed virus to the extracellular fluid, $c$ is the clearance rate of virus and $\delta$ is the clearance rate of productively infected cells. Using these parameters, the viral half-life in the respiratory tract ($t_{1/2} = \ln 2/c$) and the half-life of productively infected cells ($t_{1/2} = \ln 2/\delta$) could be estimated. Moreover, the cellular basic reproductive number (the expected number of secondary infected cells from a single productively infected cell placed in a population of susceptible cells) was calculated by

$$R_{0,c} = \frac{p\beta T_0}{c\delta}, \tag{7}$$

For initial parameterization, *Equations (4)–(6)* were simplified according to a quasi-steady state approximation (*Ikeda et al., 2016*) to

$$\frac{dT}{dt} = -\beta TV \tag{8}$$

$$\frac{dV}{dt} = rTV - \delta V, \tag{9}$$

where $r = p\beta/c$, for a form with greater numerical stability. The system of differential equations was fitted on the mean estimates of SARS-CoV-2 rVL between -2 and 10 DFSO using the entirety of individual sample data in units of copies/ml. Numerical analysis was implemented using the Fit ODE app in OriginPro 2019b (OriginLab Corporation, Northampton, MA, USA) via the Runge–Kutta method and initial parameters $V_0$, $I_0$ and $T_0$ of 4 copies/ml, 0 cells and $5 \times 10^7$ cells, respectively, for the range –5 to 10 DFSO. The analysis was first performed with *Equations (8) and (9)*. These output parameters were then used to initialize final analysis using *Equations (4)–(6)*, where the estimates for $\beta$ and $\delta$ were input as fixed and variable parameters, respectively. The fitted line and its coefficient of determination ($r^2$) were presented. The estimated half-life of SARS-CoV-2 RNA has a skewed 95% CI (*Figure 4—figure supplement 4*). As $c$ is in the denominator of the equation for half-life ($t_{1/2} = \ln 2/c$), $t_{1/2}$ is sensitive to $c$ below 1, which is the case for its lower 95% CI (*Figure 4—figure supplement 4*) and the source of the skew.

To estimate the average incubation period, we extrapolated the kinetic model to 0 $\log_{10}$ copies/ml pre-symptom onset. To estimate the average duration of shedding, we extrapolated the model to 0 $\log_{10}$ copies/ml post-symptom onset. Unlike in experimental studies, this estimate for duration of shedding was not defined by assay detection limits. To estimate the average DFSO on which SARS-CoV-2 concentration reached diagnostic levels, we extrapolated the model pre-symptom onset to the equivalent of 1 and 3 $\log_{10}$ copies/ml (chosen as example assay detection limits) in specimen concentration for NPSs immersed in 1 ml of transport media, as described by the dilution factor estimation above. The average time from respiratory infection to reach diagnostic levels was then calculated by subtracting these values from the estimated average incubation period. The extrapolated time for SARS-CoV-2 to reach diagnostic concentrations in the respiratory tract should be validated in tracing studies, in which contacts are prospectively subjected to daily sampling.

## Likelihood of respiratory particles containing virions

To calculate an unbiased estimator for viral partitioning (the expected number of viable copies in an expelled particle at a given size), we multiplied rVLs with the volume equation for spherical particles during atomization and the estimated viability proportion, according to the following equation:

$$\lambda = \frac{\pi \rho v_p \gamma \upsilon}{6} d^3, \tag{10}$$

where $\lambda$ is the expectation value, $\rho$ is the material density of the respiratory particle (997 kg/m$^3$), $v_p$ is the volumetric conversion factor (1 ml/g), $\gamma$ is the viability proportion, $\upsilon$ is the rVL and $d$ is the hydrated diameter of the particle during atomization.

The model assumed $\gamma$ was 0.1% as a population-level estimate. For influenza, approximately 0.1% of copies in particles expelled from the respiratory tract represent viable virus (*Yan et al., 2018*), which is equivalent to one viable copy in 3 $\log_{10}$ copies/ml for rVL or, after dilution in transport media, roughly one in 4 $\log_{10}$ copies/ml for specimen concentration. Respiratory specimens taken from influenza cases show positive cultures for specimen concentrations down to 4 $\log_{10}$ copies/ml (*Lau et al., 2010*). Likewise, for COVID-19 cases, recent reports also show culture-positive respiratory specimens with SARS-CoV-2 concentrations down to 4 $\log_{10}$ copies/ml (*Wölfel et al., 2020*), including from pediatric (*L'Huillier et al., 2020*) and asymptomatic (*Arons et al., 2020*) cases. Moreover, replication-competent SARS-CoV-2 has been found in respiratory specimens taken throughout the respiratory tract (mouth, nasopharynx, oropharynx and lower respiratory tract)

(*Jeong et al., 2020*; *Wölfel et al., 2020*). Taken together, these considerations suggested that the assumption for viability proportion (0.1%) was suitable to model the likelihood of respiratory particles containing viable SARS-CoV-2. In accordance with the discussion above, the model did not differentiate this population-level viability estimate based on age, symptomatology or sites of atomization. Based on the relative relationship between the residence time of expelled particles before assessment (~5 s) (*Yan et al., 2018*), we took the viability proportion to be for equilibrated particles.

Likelihood profiles were determined using Poisson statistics, as described by the probability mass function

$$P(X = k) = \frac{\lambda^k e^{-\lambda}}{k!}, \tag{11}$$

where $k$ is the number of virions partitioned within the particle. For $\lambda$, 95% CIs were determined using the variance of its rVL estimate. To determine 95% CIs for likelihood profiles from the probability mass function, we used the delta method, which specifies

$$Var(g(\theta)) \approx \sigma^2 \dot{g}(\theta)' D \dot{g}(\theta), \tag{12}$$

where $\sigma^2 \mathbf{D}$ is the covariance matrix of $\theta$ and $\dot{g}(\theta)$ is the gradient of $g(\theta)$. For the univariate Poisson distribution, $\sigma^2 \mathbf{D} = \lambda$ and

$$\dot{g}(\theta) = \frac{\lambda^{k-1} e^{-\lambda}}{k!} (k - \lambda). \tag{13}$$

## Rate profiles of particles expelled by respiratory activities

Distributions from the literature were used to determine the rate profiles of particles expelled during respiratory activities. For breathing, talking and coughing, we used data from *Johnson et al., 2011*. For singing, we used data from *Morawska et al., 2009* for smaller aerosols ($d_a$ < 20 μm) and used the profiles from talking for larger aerosols and droplets based on the oral cavity mechanism from *Johnson et al., 2011*. Rate profiles (particles/min or particles/cough) were calculated based on the corrected normalized concentration ($dC_n/d\log D_p$, in units of particles/cm$^3$) at each discrete particle size, normalization (32 size channels per decade) for the aerodynamic particle sizer used, unit conversion (cm$^3$ to l) and the sample flow rate (1 l/min). For coughing, the calculation assumed that participants coughed 10 times in the 30-s sampling interval. To determine the corrected normalized concentrations for breathing, we used a particle dilution factor of 4 and evaporation factor of 0.5, consistent with the other respiratory activities in *Johnson et al., 2011*. Breathing was taken to expel negligible quantities of larger respiratory particles based on the bronchiolar fluid film burst mechanism (*Johnson et al., 2011*). To account for intermittent breathing while talking and singing, the rate profiles for these activities included the contribution of aerosols expelled by breathing. We compared these rate profiles with those collected from talking loudly and talking quietly from *Asadi et al., 2020*. In our models, we took the diameter of dehydrated respiratory particles to be 0.3 times the initial size when atomized in the respiratory tract (*Johnson et al., 2011*; *Lieber et al., 2021*; *Liu et al., 2017b*). Equilibrium aerodynamic diameter was calculated by $d_a = d_p(\rho/\rho_0)^{1/2}$, where $d_p$ is the dehydrated diameter, $\rho$ is the material density of the respiratory particle and $\rho_0$ is the reference material density (1 g/cm$^3$). Curves based on discrete particle measurements were connected using the nonparametric Akima spline function.

## Shedding virions via respiratory droplets and aerosols

To model the respiratory shedding rate across particle size, rVL estimates and the hydrated diameters of particles expelled by a respiratory activity were input into *Equation (10)*, and the output was then multiplied by the rate profile of the activity (talking, singing, breathing or coughing). To assess the relative contribution of aerosols and droplets to mediating respiratory viral shedding for a given respiratory activity, we calculated the proportion of the cumulative hydrated volumetric rate contributed by buoyant aerosols ($d_a \leq$ 10 μm), long-range aerosols (10 μm < $d_a \leq$ 50 μm), short-range aerosols (50 μm < $d_a \leq$ 100 μm) and droplets ($d_a$ > 10 μm) for that respiratory activity. Since the Poisson mean was proportional to cumulative volumetric rate, this estimate of the relative contribution of

aerosols and droplets to respiratory viral shedding was consistent among viruses and cps in the model.

To determine the total respiratory shedding rate for a given respiratory activity across cp, we determined the cumulative hydrated volumetric rate (by summing the hydrated volumetric rates across particle sizes for that respiratory activity) of particle atomization and input it into *Equation (10)*. Using rVLs and their variances as determined by the Weibull quantile functions, we then calculated the Poisson means and their 95% CIs at the different cps.

To assess the influence of heterogeneity in rVL on individual infectiousness, we first considered transmission of A(H1N1)pdm09 via aerosols (*Cowling et al., 2013*). The 50% human infectious dose (HID$_{50}$) of aerosolized A(H1N1)pdm09 was taken to be 1–3 virions (*Fabian et al., 2008*). To determine the expected time required for a A(H1N1)pdm09 case to shed one virion via aerosols, we took the reciprocal of the Poisson means and their 95% CIs at the different cps of the estimated shedding rates. The expected time required for a COVID-19 case to shed one virion via aerosols or one virion via droplets or aerosols was determined in a same manner.

## Data availability

The systematic dataset and model outputs from this study were uploaded to Zenodo (https://zenodo.org/record/4658971). The code generated during this study is available at GitHub (https://github.com/paulzchen/sars2-heterogeneity; *Chen, 2020*; copy archived at swh:1:rev:06649ccfb6e92918b439332314ebf330abfa3d16). The systematic review protocol was prospectively registered on PROSPERO (registration number, CRD42020204637).

# Acknowledgements

We thank T Alba (Toronto) for discussion on statistical methods. We thank J Jimenez (Colorado) for discussion on the characteristics of aerosols and droplets. We thank E Lavezzo and A Chrisanti (Padova) and A Wyllie, A Ko and N Grubaugh (Yale) for responses to data inquiries. PZC was supported by the NSERC Vanier Scholarship (608544). DNF was supported by the Canadian Institutes of Health Research (Canadian COVID-19 Rapid Research Fund, OV4-170360). FXG was supported by the NSERC Senior Industrial Research Chair program, NSERC Discovery Grant program and the Toronto COVID-19 Action Fund.

# Additional information

### Funding

| Funder | Grant reference number | Author |
|---|---|---|
| Natural Sciences and Engineering Research Council of Canada | Vanier Scholarship 608544 | Paul Z Chen |
| Canadian Institutes of Health Research | Canadian COVID-19 Rapid Research Fund OV4-170360 | David N Fisman |
| Natural Sciences and Engineering Research Council of Canada | Senior Industrial Research Chair | Frank X Gu |
| Toronto COVID-19 Action Fund | | Frank X Gu |

The funders had no role in study design, data collection and interpretation, or the decision to submit the work for publication.

### Author contributions

Paul Z Chen, Conceptualization, Formal analysis, Investigation, Visualization, Methodology, Writing - original draft, Writing - review and editing; Niklas Bobrovitz, Investigation, Methodology, Writing - review and editing; Zahra Premji, Resources, Methodology, Writing - review and editing; Marion

Koopmans, David N Fisman, Supervision, Writing - review and editing; Frank X Gu, Supervision, Funding acquisition, Writing - review and editing

## Author ORCIDs

Paul Z Chen ⓘ https://orcid.org/0000-0001-5261-1610
Niklas Bobrovitz ⓘ https://orcid.org/0000-0001-7883-4484
Zahra Premji ⓘ http://orcid.org/0000-0002-6899-0528
David N Fisman ⓘ https://orcid.org/0000-0001-5009-6926
Frank X Gu ⓘ https://orcid.org/0000-0001-8749-9075

## Decision letter and Author response

Decision letter https://doi.org/10.7554/eLife.65774.sa1
Author response https://doi.org/10.7554/eLife.65774.sa2

## Additional files

### Supplementary files

• Transparent reporting form

### Data availability

The systematic dataset and model outputs from this study are uploaded to Zenodo (https://zenodo.org/record/4658971). The code generated during this study is available at GitHub (https://github.com/paulzchen/sars2-heterogeneity; copy archived at https://archive.softwareheritage.org/swh:1:rev:06649ccfb6e92918b439332314ebf330abfa3d16). The systematic review protocol was prospectively registered on PROSPERO (registration number, CRD42020204637).

The following dataset was generated:

| Author(s) | Year | Dataset title | Dataset URL | Database and Identifier |
|---|---|---|---|---|
| Chen PZ, Bobrovitz N, Premji Z, Koopmans M, Fisman DN, Gu FX | 2020 | Heterogeneity in transmissibility and shedding SARS-CoV-2 via droplets and aerosols | https://zenodo.org/record/4658971 | Zenodo, 10.5281/zenodo/4658971 |

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

# Appendix 1

**Appendix 1—table 1.** Characteristics of contributing studies.

| Study* | Country | No. of cases included (no. of specimens) | No. of pediatric cases (no. of specimens) | No. of asymptomatic cases (no. of specimens) | Disease caused by virus (WHO case definition) | Treatments given (type)† | Individual data extracted (diluent volume reported)‡ | Adjusted viral load§ (type of specimen) | Weight, % (meta-analysis category)‖ | Weight, % (meta-regression) | Risk of bias¶ |
|---|---|---|---|---|---|---|---|---|---|---|---|
| *Argyropoulos et al., 2020* | USA | 205 (205) | 0 | 0 | COVID-19 (confirmed) | N/A | No (no) | Yes (NPS) | 3.80 (V), 5.09 (A), 3.98 (S/Ps) | 2.13 | ******** |
| *Baggio et al., 2020* | Switzerland | 405 (405) | 58 (58) | 0 | COVID-19 (confirmed) | N/A | Yes (no) | Yes (NPS) | 3.84 (V), 5.14 (A), 13.9 (P), 3.88 (S/Ps) | 4.20 | ******* |
| *Fajnzylber et al., 2020* | USA | - (31) | 0 | 0 | COVID-19 (confirmed) | Yes (remdesivir) | Yes (yes) | Yes (NPS, OPS) No (Spu) | 3.45 (V), 4.62 (A), 3.76 (S/Ps) | 0.32 | ******** |
| *Hung et al., 2020* | South Korea | 2 (8) | 1 (6) | 0 | COVID-19 (confirmed) | N/A | Yes (no) | Yes (NPS, OPS) | 2.53 (V), 4.06 (S/Ps) | 0.08 | ****** |
| *Hung et al., 2020* | South Korea | 12 (27) | 12 (27) | 3 (7) | COVID-19 (confirmed) | N/A | Yes (no) | Yes (NPS) | 3.43 (V), 17.5 (P), 4.10 (S/Ps), 14.3 (As) | 0.28 | ******** |
| *Hung et al., 2020* | China | 41 (310) | 0 | 0 | COVID-19 (confirmed) | Yes (control group: lopinavir ritonavir, antimicrobial treatment for secondary bacterial infection as indicated clinically, hydrocortisone for those requiring oxygen support) | No (no) | Yes (NPS, OPS, POS) | 3.81 (V), 5.10 (A), 3.81 (S/Ps) | 3.22 | ********* |
| *Hurst et al., 2020* | USA | 133 (133) | 54 (54) | 52 (52) | COVID-19 (confirmed) | Yes (remdesivir) | Yes (no) | Yes (NPS) | 3.77 (V), 15.3 (P), 3.88 (S/Ps), 21.6 (As) | 1.38 | ******** |
| *Iwasaki et al., 2020* | Japan | 5 (5) | 0 | 0 | COVID-19 (confirmed) | N/A | Yes (no) | Yes (NPS) | 2.53 (V), 3.37 (A), 4.12 (S/Ps) | 0.05 | **** |
| *Kawasuji et al., 2020* | Japan | 16 (16) | - | - | COVID-19 (confirmed) | Yes (antivirals, antibiotics – specifics not reported) | Yes (no) | Yes (NPS) | 3.15 (V) | 0.18 | ***** |
| *L'Huillier et al., 2020* | Switzerland | 23 (23) | 23 (23) | 0 | COVID-19 (confirmed) | N/A | Yes (no) | Yes (NPS) | 2.91 (V), 14.7 (P), 3.73 (S/Ps) | 0.24 | ******** |
| *Lavezzo et al., 2020* | Italy | 103 (110) | 2 (3) | 49 (49) | COVID-19 (confirmed) | N/A | Yes (yes) | Yes (NPS, OPS) | 3.77 (V), 5.03 (A), 11.57 (P), 3.57 (S/Ps), 21.8 (As) | 1.14 | ******* |
| *Lennon et al., 2020* | USA | 2200 (2,200) | 18 (18) | 2200 (2200#) | COVID-19 (confirmed) | N/A | No (yes) | Yes (NPS) | 3.88 (V), 5.20 (A), 24.0 (As) | 22.84 | ********* |

*Continued on next page*

*Appendix 1—table 1 continued*

| Study* | Country | No. of cases included (no. of specimens) | No. of pediatric cases (no. of specimens) | No. of asymptomatic cases (no. of specimens) | Disease caused by virus (WHO case definition) | Treatments given (type)† | Individual data extracted (diluent volume reported)‡ | Adjusted viral load§ (type of specimen) | Weight, % (meta-analysis category)ǁ | Weight, % (meta-regression) | Risk of bias¶ |
|---|---|---|---|---|---|---|---|---|---|---|---|
| Lucas et al., 2020 | USA | 24 (33) | 0 | 0 | COVID-19 (confirmed) | Yes (tocilizumab for moderate and severe patients, glucocorticoid and vasopressor for severe patients) | Yes (yes) | Yes (NPS) | 3.51 (V), 4.69 (A), 4.08 (S/Ps) | 0.34 | ******* |
| Mitjà et al., 2020 | Spain | 148 (296) | 0 | 0 | COVID-19 (confirmed) | N/A | No (no) | Yes (NPS) | 3.81 (V), 5.10 (A), 3.93 (S/Ps) | 3.07 | ********* |
| Pan et al., 2020 | China | 75 (104) | - | 0 | COVID-19 (confirmed) | N/A | Yes (no) | Yes (OPS) No (Spu) | 3.45 (V), 2.50 (S/Ps) | 1.24 | **** |
| Peng et al., 2020 | China | 6 (6) | 0 | 0 | COVID-19 (confirmed) | Yes (arbidol, lopinavir, ritonavir, interferon alfa-2b inhalation) | Yes (no) | Yes (OPS) | 3.03 (V), 4.05 (A), 4.02 (S/Ps) | 0.06 | ******** |
| Perera et al., 2020 | China | - (36) | 0 | - | COVID-19 (confirmed) | Yes lopinavir-ritonavir alone, combination lopinavir-ritonavir and ribavirin, ribavirin and β interferon, β interferon alone, combination ribavirin, β interferon, and tocilizumab, and corticosteroid | Yes (no) | Yes (NPA, NPS, OPS, Spu) | 3.23 (V), 4.32 (A) | 0.39 | **** |
| Shi et al., 2020 | China | 103 (103) | 0 | 0 | COVID-19 (confirmed) | No (samples drawn before antivirals given) | Yes (no) | Yes (NPS, OPS) | 3.87 (V), 5.18 (A), 4.34 (S/Ps) | 1.07 | ***** |
| Shrestha et al., 2020 | USA | 171 (171) | 0 | 0 | COVID-19 (confirmed) | Yes (indicated no hydroxychloroquine or other COVID-19-related treatments were used) | Yes (no) | Yes (NPS) | 3.79 (V), 5.07 (A), 3.86 (S/Ps) | 1.78 | ******* |
| To et al., 2020 | China | 23 (51) | 0 | 0 | COVID-19 (confirmed) | N/A | Yes (yes) | Yes (ETA, POS) | 3.37 (V), 4.51 (A), 3.25 (S/Ps) | 0.53 | ********* |
| van Kampen et al., 2021 | The Netherlands | - (154) | 0 | 0 | COVID-19 (confirmed) | Yes (lopinavir-ritonavir with or without ribavirin and/or interferon beta 1b) | Yes (yes) | Yes (NPS, Spu) | 3.80 (V), 5.09 (A), 4.10 (S/Ps) | 1.60 | ******** |
| Vetter et al., 2020 | Switzerland | 5 (63) | 0 | 0 | COVID-19 (confirmed) | Yes (paracetamol, alfuzosin, ibuprofen, enoxaparin, amoxicillin clarithromycin, piperacillin, tazobactam, lopinavir, ritonavir, folic acid) | Yes (yes) | Yes (NPS, OPS) | 3.68 (V), 4.93 (A), 4.14 (S/Ps) | 0.65 | ********* |
| Wölfel et al., 2020 | Germany | 9 (136) | 0 | 1 (4) | COVID-19 (confirmed) | N/A | Yes (yes) | Yes (NPS, OPS) No (Spu) | 3.76 (V), 5.03 (A), 3.93 (S/Ps) | 1.38 | ******* |

*Continued on next page*

*Appendix 1—table 1 continued*

| Study* | Country | No. of cases included (no. of specimens) | No. of pediatric cases (no. of specimens) | No. of asymptomatic cases (no. of specimens) | Disease caused by virus (WHO case definition) | Treatments given (type)† | Individual data extracted (diluent volume reported)‡ | Adjusted viral load§ (type of specimen) | Weight, % (meta-analysis category)‖ | Weight, % (meta-regression) | Risk of bias¶ |
|---|---|---|---|---|---|---|---|---|---|---|---|
| Wyllie et al., 2020 | USA | 40 (42) | - | 9 (9) | COVID-19 (confirmed) | N/A | Yes (yes) | Yes (NPS) | 3.55 (V), 4.75 (A), 4.00 (S/Ps), 18.3 (As) | 0.44 | ******* |
| Xu et al., 2020 | China | 7 (14) | 7 (14) | 1 (1) | COVID-19 (confirmed) | Yes (α-interferon oral spray, azithromycin) | Yes (no) | Yes (NPS) | 3.40 (V). 17.3 (P), 4.1 (S/Ps) | 0.15 | ******** |
| Yonker et al., 2020 | USA | 17 (17) | 14 (14) | 0 | COVID-19 (confirmed) | N/A | Yes (no) | Yes (NPS) | 2.58 (V), 9.79 (P), 3.28 (S/Ps) | 0.18 | ****** |
| Zhang et al., 2020b | China | 9 (9) | 0 | 0 | COVID-19 (confirmed) | N/A | Yes (no) | Yes (NPS, OPS) | 2.97 (V), 3.97 (A), 3.68 (S/Ps) | 0.09 | ******** |
| Zheng et al., 2020 | China | - (19) | 0 | 0 | COVID-19 (confirmed) | Yes (gamma globulin, glucocorticoids, antibiotics, antiviral combination of interferon α inhalation, lopinavir-ritonavir combination, arbidol, favipiravir, and darunavir-cobicistat) | Yes (no) | Yes (POS, Spu) | 3.66 (V), 4.90 (A), 4.23 (S/Ps) | 0.20 | ******* |
| Zou et al., 2020 | China | 14 (55) | 0 | 1 (4) | COVID-19 (confirmed) | N/A | Yes (no) | Yes (NPS, OPS) | 3.64 (V), 4.87 (A), 3.65 (S/Ps) | 0.57 | ******* |
| Chen et al., 2006 | China | 154 (154#) | 0 | 0 | SARS (confirmed) | N/A | Yes (no) | Yes (NPS) | 14.0 (V) | 1.59 | ******** |
| Chu et al., 2004** | China | 11 (11) | 0 | 0 | SARS (confirmed) | Yes (control group: ribavirin, hydrocortisone, methylprednisolone) | Yes (yes) | Yes (NPS) | 8.6 (V) | 0.11 | ********* |
| Chu et al., 2005 | China | 57 (57) | 0 | 0 | SARS (confirmed) | N/A | Yes (yes) | Yes (NPA) | 13.3 (V) | 0.59 | ********* |
| Cheng et al., 2004 | China | 59 (59) | 0 | 0 | SARS (confirmed) | Yes (amoxicillin-clavulanate, azithromycin, levofloxacin, ribavirin, hydrocortisone, prednisolone, methylprednisolone) | Yes (yes) | Yes (NPA) | 13.4 (V) | 0.61 | ********* |
| Hung et al., 2004 | China | 60 (60) | 0 | 0 | SARS (confirmed) | N/A | No (yes) | Yes (NPA) | 13.5 (V) | 0.62 | ******* |
| Peiris et al., 2003** | China | 14 (42) | 0 | 0 | SARS (confirmed) | Yes (ribavirin, hydrocortisone, prednisolone, methylprednisolone) | Yes (no) | Yes (NPA) | 13.4 (V) | 0.44 | ******** |
| Poon et al., 2003 | China | 40 (40) | 0 | 0 | SARS (confirmed) | N/A | No (yes) | Yes (NPA) | 11.3 (V) | 0.42 | ***** |
| Poon et al., 2004 | China | - (43) | 0 | 0 | SARS (confirmed) | N/A | No (yes) | Yes (NPA) | 12.5 (V) | 0.45 | ******* |

*Appendix 1—table 1 continued*

| Study* | Country | No. of cases included (no. of specimens) | No. of pediatric cases (no. of specimens) | No. of asymptomatic cases (no. of specimens) | Disease caused by virus (WHO case definition) | Treatments given (type)† | Individual data extracted (diluent volume reported)‡ | Adjusted viral load§ (type of specimen) | Weight, % (meta-analysis category)‖ | Weight, % (meta-regression) | Risk of bias¶ |
|---|---|---|---|---|---|---|---|---|---|---|---|
| *Rodrigues Guimarães Alves et al., 2020* | Brazil | 86 (86) | - | 15 (15) | A(H1N1)pdm09 (confirmed) | Yes (oseltamivir) | No (yes) | Yes (NPA, NPS, OPS) | 3.7 (V) | 0.89 | ***** |
| *Chan et al., 2011* | China | 58 (58) | 0 | 0 | A(H1N1)pdm09 (confirmed) | Yes (oseltamivir, zanamivir, peramivir) | Yes (no) | Yes (NPA, NPS, OPS) | 3.7 (V) | 0.60 | ****** |
| *Cheng et al., 2010* | China | 60 (60) | - | 0 | A(H1N1)pdm09 (confirmed) | No (pretreatment samples) | No (no) | Yes (NPA) | 3.7 (V) | 0.62 | ****** |
| *Cowling et al., 2010* | China | 45 (54) | 22 (31) | 0 | A(H1N1)pdm09 (confirmed) | Yes (oseltamivir) | Yes (yes) | Yes (NPS, OPS) | 3.7 (V) | 0.56 | ********* |
| *Duchamp et al., 2010* | France | 209 (209) | 209 (209) | 0 | A(H1N1)pdm09 (confirmed) | Yes (oseltamivir, zanamivir) | No (yes) | Yes (NPS) | 3.8 (V) | 2.17 | ***** |
| *Esposito et al., 2011* | Italy | 74 (282) | 74 (282) | 0 | A(H1N1)pdm09 (confirmed) | No | Yes (yes) | Yes (NPS) | 3.8 (V) | 2.93 | ******* |
| *Hung et al., 2010* | China | 87 (87) | - | 0 | A(H1N1)pdm09 (confirmed) | Yes (oseltamivir) | Yes (no) | Yes (NPA, NPS) | 3.8 (V) | 0.90 | ****** |
| *Ip et al., 2016* | China | 17 (20) | 7 (-) | 0 | A(H1N1)pdm09 (confirmed) | N/A | Yes (no) | Yes (NPS, OPS) | 3.6 (V) | 0.21 | ******* |
| *Ito et al., 2012* | Japan | 34 (34) | - | 0 | A(H1N1)pdm09 (confirmed) | No (pretreatment samples) | Yes (yes) | Yes (NPS) | 3.7 (V) | 0.35 | ***** |
| *Killingley et al., 2010* | United Kingdom | 12 (21) | - | 0 | A(H1N1)pdm09 (confirmed) | Yes (oseltamivir) | Yes (yes) | Yes (NPS) | 3.5 (V) | 0.22 | ******** |
| *Launes et al., 2012* | Spain | 47 (47) | 47 (47) | 0 | A(H1N1)pdm09 (confirmed) | No (pretreatment samples) | No (no) | Yes (NPA) | 3.7 (V) | 0.49 | ******* |
| *Lee et al., 2011a* | China | 48 (48) | 0 | 0 | A(H1N1)pdm09 (confirmed) | No (pretreatment samples) | No (no) | Yes (NPA) | 3.7 (V) | 0.50 | ******** |
| *Lee et al., 2011a* | Singapore | 578 (578) | 231 (231) | 0 | A(H1N1)pdm09 (confirmed) | No (pretreatment samples) | No (no) | Yes (NPS) | 3.8 (V) | 6.00 | ********* |
| *Li et al., 2010a* | China | 581 (581) | 522 (522) | 0 | A(H1N1)pdm09 (confirmed) | Yes (oseltamivir) | No (no) | Yes (OPS) | 3.8 (V) | 6.03 | ******** |
| *Li et al., 2010b* | China | 27 (59) | - | 0 | A(H1N1)pdm09 (confirmed) | No (control group no treatment) | No (no) | Yes (NPA, NPS, OPS) | 3.7 (V) | 0.61 | ******* |
| *Loeb et al., 2012* | Canada | 97 (218) | - | - (17) | A(H1N1)pdm09 (confirmed) | No | No (no) | Yes (NPS) | 3.8 (V) | 2.26 | ******* |
| *Lu et al., 2012* | China | 13 (25) | - | 0 | A(H1N1)pdm09 (confirmed) | Yes (oseltamivir, zanamivir) | Yes (no) | Yes (NPS) | 3.5 (V) | 0.26 | ******* |

*Continued on next page*

*Appendix 1—table 1 continued*

| Study* | Country | No. of cases included (no. of specimens) | No. of pediatric cases (no. of specimens) | No. of asymptomatic cases (no. of specimens) | Disease caused by virus (WHO case definition) | Treatments given (type)† | Individual data extracted (diluent volume reported)‡ | Adjusted viral load§ (type of specimen) | Weight, % (meta-analysis category)‖ | Weight, % (meta-regression) | Risk of bias¶ |
|---|---|---|---|---|---|---|---|---|---|---|---|
| *Meschi et al., 2011* | Italy | 533 (533) | 0 | 0 | A(H1N1) pdm09 (confirmed) | No (pretreatment samples) | No (no) | Yes (NPS) | 3.8 (V) | 0.92 | ********* |
| *Ngaosuwankul et al., 2010* | China | 12 (33) | - | 0 | A(H1N1) pdm09 (confirmed) | No (pretreatment samples) | No (yes) | Yes (NPA, NPS, OPS) | 3.6 (V) | 0.34 | ****** |
| *Rath et al., 2012* | Germany | 27 (41) | 27 (41) | 0 | A(H1N1) pdm09 (confirmed) | Yes (oseltamivir) | Yes (yes) | Yes (NPS) | 3.7 (V) | 0.43 | ********* |
| *Suess et al., 2010* | Germany | 51 (129) | 12 (-) | 1 (1) | A(H1N1) pdm09 (confirmed) | Yes (oseltamivir) | No (no) | Yes (NPA, NPS, OPS) | 3.8 (V) | 1.34 | ******** |
| *Thai et al., 2014* | Vietnam | 33 (123) | 16 (-) | 5 (28) | A(H1N1) pdm09 (confirmed) | Yes (oseltamivir) | Yes (yes) | Yes (NPS) | 3.8 (V) | 1.28 | ********* |
| *To et al., 2010a* | China | 50 (50) | 0 | 0 | A(H1N1) pdm09 (confirmed) | Yes (oseltamivir, zanamivir, inotropes) | No (no) | Yes (NPA, NPS) | 3.6 (V) | 0.52 | ****** |
| *To et al., 2010b* | China | 22 (22) | - | 0 | A(H1N1) pdm09 (confirmed) | No (pretreatment samples) | No (no) | Yes (NPA, NPS, OPS) | 3.4 (V) | 0.23 | ***** |
| *Watanabe et al., 2011* | Japan | 251 (251) | 251 (251) | 0 | A(H1N1) pdm09 (confirmed) | No (pretreatment samples) | No (yes) | Yes (NPA) | 3.8 (V) | 2.61 | ********** |
| *Wu et al., 2012* | China | 64 (89) | - | 0 | A(H1N1) pdm09 (confirmed) | Yes (oseltamivir) | No (yes) | Yes (NPS) | 3.7 (V) | 5.53 | ******* |
| *Yang et al., 2011* | China | 251 (251) | - | 0 | A(H1N1) pdm09 (confirmed) | N/A | No (yes) | Yes (OPS) | 3.8 (V) | 6.57 | ***** |

*Data shown as '-' were not obtained from the paper or authors.

†Responses of 'N/A' indicate that no details were reported on treatment for COVID-19 in the study.

‡For studies reporting specimen measurements as individual sample data (either in numerical or graphical formats), the sample data was extracted for analysis.

§Specimen measurements were converted to rVLs based on the dilution factor for specimens in transport media.

‖Abbreviations for random-effects meta-analyses: virus meta-analysis (V), adult subgroup (A), pediatric subgroup (P), symptomatic/presymptomatic subgroup (S/Ps) and asymptomatic subgroup (As).

¶The hybrid JBI critical appraisal checklist was used, with more stars indicating lower risk of bias. Studies were considered to have low risk of bias if they met the majority of the items (≥6/10 items). Results from each study are shown in *Appendix 1—table 2*.

#For these studies, 2147 (*Lennon et al., 2020*) and 134 (*Chen et al., 2006*) individual specimen measurements were obtained for the individual sample datasets.

**For *Chu et al., 2004*, only specimen measurements at 20 DFSO were collected as 5–15 DFSO were specimens reported in *Peiris et al., 2003*.

NPS: nasopharyngeal swab; OPS, oropharyngeal swab; Spu: sputum; POS, posterior oropharyngeal saliva; NPA: nasopharyngeal aspirate; ETA: endotracheal aspirate; DFSO: days from symptom onset; rVL: respiratory viral load.

**Appendix 1—table 2.** Assessment of risk of bias based on the hybrid JBI critical appraisal checklist.

| Study | Checklist items* | | | | | | | | | |
|---|---|---|---|---|---|---|---|---|---|---|
| | 1 | 2 | 3 | 4 | 5 | 6 | 7 | 8 | 9 | 10 |
| Argyropoulos et al., 2020 | Y | Y | Y | Y | N | Y | Y | N | Y | Y |
| Baggio et al., 2020 | Y | Y | Y | Y | N | Y | Y | N | N | Y |
| Fajnzylber et al., 2020 | Y | Y | Y | Y | N | Y | Y | Y | N | Y |
| Hung et al., 2020 | N | Y | N | Y | N | Y | Y | N | Y | Y |
| Hung et al., 2020 | Y | Y | Y | Y | N | Y | Y | N | Y | Y |
| Hung et al., 2020 | Y | Y | Y | Y | Y | Y | Y | N | Y | Y |
| Hung et al., 2020 | Y | Y | Y | Y | N | Y | Y | N | Y | Y |
| Iwasaki et al., 2020 | Y | N | U | U | N | Y | Y | N | N | Y |
| Kawasuji et al., 2020 | Y | Y | U | U | N | Y | Y | N | N | Y |
| L'Huillier et al., 2020 | Y | Y | Y | Y | N | Y | Y | N | Y | Y |
| Lavezzo et al., 2020 | Y | Y | N | Y | N | Y | Y | N | Y | Y |
| Lennon et al., 2020 | Y | Y | Y | Y | Y | Y | Y | N | Y | Y |
| Lucas et al., 2020 | Y | Y | U | U | N | Y | Y | Y | Y | Y |
| Mitjà et al., 2020 | Y | Y | Y | Y | Y | Y | Y | N | Y | Y |
| Pan et al., 2020 | Y | N | U | U | N | Y | Y | N | N | Y |
| Peng et al., 2020 | Y | Y | Y | Y | N | Y | Y | N | Y | Y |
| Perera et al., 2020 | Y | N | U | U | N | Y | Y | N | N | Y |
| Shi et al., 2020 | Y | Y | U | U | N | Y | Y | N | N | Y |
| Shrestha et al., 2020 | N | Y | Y | Y | N | Y | Y | N | Y | Y |
| To et al., 2020 | Y | Y | Y | Y | N | Y | Y | Y | Y | Y |
| van Kampen et al., 2021 | Y | Y | Y | Y | N | Y | Y | N | Y | Y |
| Vetter et al., 2020 | Y | Y | Y | Y | N | Y | Y | Y | Y | Y |
| Wölfel et al., 2020 | Y | Y | N | U | N | Y | Y | Y | Y | Y |
| Wyllie et al., 2020 | Y | Y | U | Y | N | Y | Y | Y | N | Y |
| Xu et al., 2020 | Y | Y | Y | Y | N | Y | Y | N | Y | Y |
| Yonker et al., 2020 | Y | Y | N | U | N | Y | Y | N | Y | Y |
| Zhang et al., 2020b | Y | Y | Y | Y | N | Y | Y | N | Y | Y |
| Zheng et al., 2020 | Y | Y | Y | Y | N | Y | Y | N | N | Y |
| Zou et al., 2020 | N | Y | Y | Y | N | Y | Y | N | Y | Y |
| Chen et al., 2006 | Y | Y | Y | Y | N | Y | Y | N | Y | Y |
| Chu et al., 2004 | Y | Y | Y | Y | N | Y | Y | Y | Y | Y |
| Chu et al., 2005 | Y | Y | Y | Y | N | Y | Y | Y | Y | Y |
| Cheng et al., 2004 | Y | Y | Y | Y | N | Y | Y | Y | Y | Y |
| Hung et al., 2004 | Y | Y | U | U | N | Y | Y | Y | Y | Y |
| Peiris et al., 2003 | Y | Y | Y | Y | N | Y | Y | N | Y | Y |
| Poon et al., 2003 | Y | N | U | U | N | Y | Y | Y | N | Y |
| Poon et al., 2004 | Y | N | Y | Y | N | Y | Y | Y | N | Y |
| Rodrigues Guimarães Alves et al., 2020 | Y | N | U | U | N | Y | Y | Y | N | Y |
| Chan et al., 2011 | Y | Y | U | U | N | Y | Y | N | Y | Y |
| Cheng et al., 2010 | Y | N | Y | Y | N | Y | Y | N | N | Y |
| Cowling et al., 2010 | Y | Y | Y | Y | N | Y | Y | Y | Y | Y |

*Continued on next page*

*Appendix 1—table 2 continued*

**Checklist items\***

| | | | | | | | | | | |
|---|---|---|---|---|---|---|---|---|---|---|
| *Duchamp et al., 2010* | Y | Y | U | U | N | Y | Y | Y | N | N |
| *Esposito et al., 2011* | Y | Y | Y | U | N | Y | Y | Y | N | Y |
| *Hung et al., 2010* | N | Y | Y | Y | N | Y | Y | N | N | Y |
| *Ip et al., 2016* | Y | Y | Y | U | N | Y | Y | N | Y | Y |
| *Ito et al., 2012* | Y | N | U | U | N | Y | Y | Y | N | Y |
| *Killingley et al., 2010* | Y | Y | Y | Y | N | Y | Y | Y | N | Y |
| *Launes et al., 2012* | Y | Y | Y | Y | N | Y | Y | N | N | Y |
| *Lee et al., 2011a* | Y | Y | Y | Y | N | Y | Y | N | Y | Y |
| *Lee et al., 2011a* | Y | Y | Y | Y | Y | Y | Y | N | Y | Y |
| *Li et al., 2010a* | Y | Y | Y | Y | N | Y | Y | N | Y | Y |
| *Li et al., 2010b* | Y | Y | Y | Y | N | Y | Y | N | N | Y |
| *Loeb et al., 2012* | Y | Y | Y | Y | N | Y | Y | N | N | Y |
| *Lu et al., 2012* | Y | Y | Y | Y | N | Y | Y | N | N | Y |
| *Meschi et al., 2011* | Y | Y | Y | Y | Y | Y | Y | N | Y | Y |
| *Ngaosuwankul et al., 2010* | Y | Y | U | U | N | Y | Y | Y | N | Y |
| *Rath et al., 2012* | Y | Y | Y | Y | N | Y | Y | Y | Y | Y |
| *Suess et al., 2010* | Y | Y | Y | Y | N | Y | Y | N | Y | Y |
| *Thai et al., 2014* | Y | Y | Y | Y | N | Y | Y | Y | Y | Y |
| *To et al., 2010a* | Y | Y | U | U | N | Y | Y | N | Y | Y |
| *To et al., 2010b* | Y | Y | U | U | N | Y | Y | N | N | Y |
| *Watanabe et al., 2011* | Y | Y | Y | Y | Y | Y | Y | Y | Y | Y |
| *Wu et al., 2012* | Y | Y | Y | U | N | Y | Y | Y | N | Y |
| *Yang et al., 2011* | Y | N | U | U | Y | Y | Y | N | N | Y |

\*Descriptions of each item are included in the hybrid JBI critical appraisal checklist (Appendix). Y (green), U (yellow) and N (red) represent yes, unclear and no, respectively.

