## [Decision Letter]

**Acceptance summary:**

The authors performed a systematic literature review and meta-analysis to develop a dataset of respiratory viral loads (rVLs)for three viruses (SARS-CoV-2, SARS-CoV-1 and influenza A(H1N1)pdm09). Furthermore, the kinetics of viral shedding over time during a respiratory infection are studied, and a model is developed for infectiousness via shedding of viable virus in aerosols and droplets. The study appears robust and comprehensive, and the results are valuable and contribute to the scientific knowledge in this field.

**Decision letter after peer review:**

Thank you for submitting your article "Heterogeneity in transmissibility and shedding SARS-CoV-2 via droplets and aerosols" for consideration by *eLife*. Your article has been reviewed by 2 peer reviewers, and the evaluation has been overseen by a Senior/Reviewing Editor. The following individual involved in review of your submission has agreed to reveal their identity: Lucie Vermeulen (Reviewer #1).

Summary:

This is a very interesting study on an important subject. It uses a combination of approaches (systematic review, meta-regression, mathematical modelling) to study the association between the variability of respiratory viral loads (rVL) and heterogeneity in transmission rates. The authors argue that variability of rVL is a main determinant of the high heterogeneity of transmission rates and translate the rVL distribution into transmission probabilities for different transmission modes (droplets, aerosols; breathing, speaking, singing). These conclusions are interesting and potentially relevant for public health. The combination of rVL data from >60 studies represents an impressive amount of work which may also be useful for future research.

The paper does not stop at a descriptive summary of these data but uses several modelling approaches (meta-regression, "translation" into transmission probabilities, dynamical modelling) to interpret these data. The evidence provided by these analyses is more tentative than presented by the authors in this version of the manuscript.

Essential revisions:

1. The meta-regression is based on only three viral species and hence it is unclear how generalisable the observed association between rVL and transmission heterogeneity is. In the best case the data show that the three virus species exhibit significantly different rVL-variation, which coincides with their different k values at the epidemiological level. However, this latter association is essentially based on only three data points (i.e. the three viral species). The current meta-regression approach (applying a simple linear regression but essentially ignoring the fact that all studies stem from only three virus species; i.e., ignoring the hierarchical nature of the data) provides p values which strongly exaggerate the degree of evidence.

2. One major potential confounder is the very strong dependence of rVL on infection time. The authors consider this in the section "SARS-CoV 2 kinetics during respiratory infection" where they show also a substantial variation of rVL across different strata of days from symptom onset (DFSO). However, it is unclear to what extent this affects the previous analyses (e.g., the meta-regression models). More fundamentally, the cross-sectional nature of the rVL data leads almost by necessity to an overestimation of the variability in the transmission potential of infections and in a very strong dependency of the rVL variation on the distribution of sampling time. Even a stratification on DFSO can only partly address these problems, firstly because DFSO is in most cases associated with a substantial uncertainty which in the case of a highly dynamic infection will translate into an even larger variation of rVLs. Moreover, even if DFSO were an exact measure of infection time, different infections do not need to be synchronous (e.g. because of stochastic effects or variation of the processes corresponding to the model parameters across individuals) such that different individuals will have their peak rVL at different time points. Taken together this implies that a reliable measure of heterogeneity would require determining something like the area under the curve of rVL (which of course is very challenging).

3. This limitation also strongly affects the practical, public health relevance of the findings. For example, the authors state that "Our analyses suggest that heterogeneity in rVL may be generally associated with over-dispersion for viral respiratory infections. In this case, rVL distribution can serve as an early correlate for transmission patterns, including super-spreading, during outbreaks of novel respiratory viruses, providing insight for disease control before large-scale epidemiological studies empirically characterise k ". This potential application assumes that the timing of rVL measurements is known early in a pandemic and that it can be controlled for, which requires a detailed knowledge of the within patient dynamics of the virus. I would assume that achieving this knowledge would take at least as long as estimating k in epidemiological studies. Thus it may be more appropriate to think about the two approaches in characterising heterogeneity as complementary (in the context of epidemiological triangulation; i.e. both approaches having their weaknesses and biases but which can be overcome in a joint consideration; generally, I think that attempting to achieve such a triangulation is one of the main strengths of the present study, despite its limitations).

4. The variation of rVL might also be strongly driven by the sampling method/procedure (even the same method will give very different results across health-care workers), which implies the same problems as (2) – i.e. overestimation of rVL variability and potential confounding.

5. The authors note that "Talking, singing and coughing expelled virions at greater proportions via droplets (80.6-86.0%) than aerosols (14.0-19.4%)." It should be noted that although more virions are expelled via droplets than aerosols according to the findings of this study, exposure to droplets and aerosols is not equal and this could affect the probability of transmission via these routes. For example, if social distancing and masking is observed then it is possible that larger droplets are more easily captured by masks or fall on the ground quickly and do not reach a susceptible individual, while aerosols do. Furthermore, smaller droplets and aerosols can penetrate more deeply into the lungs. It is as of yet unclear whether this would influence the probability of becoming (more severely) infected. This may also differ per virus. A discussion on these issues is relevant.

6. The authors also note that their results "support aerosol spread as a transmission mode for SARS-CoV-2, including for conditional superspreading by highly infectious cases. However, with short durations of stay in well-ventilated areas, the exposure risk for aerosols, including long-range and buoyant ones, remains correlated with proximity to infectious cases."

7. A methodological note on the modelling that may affect the results (but likely do not impact the conclusions strongly) is the following.

The authors take a value of 0.1% for the fraction of SARS-CoV-2 RNA copies that represents viable virus (parameter 𝛾). This value is quite uncertain. More literature (not yet peer-reviewed) exists on the fraction of SARS-CoV-2 RNA copies that is infectious virus, providing different values. Van Kampen et al. (2020) only found a cytopathic effect on Vero cells if the swab sample from patients contained more than 7 log_10_ RNA copies/mL. Fears et al. (2020) find an average of 0.003 (range 0.0008 – 0.02) CCID50/RNA copy. However, Lednicky et al. (2020) sampled SARS-CoV-2-containing aerosols in a room with COVID-19 patients with air samplers using a water vapor condensation mechanism and as such collect virus particles without damaging them, and found an average of 0.6 CCID50/RNA copy, much higher. The model is likely sensitive to this parameter, and this could influence the result. These issues should be taken into account in the revision.

8. Another methodological point is that more datasets in literature are available on the emission rates and size distribution of particles during breathing, speaking, coughing etc., than are currently used to base the model on. Schijven et al. (2020) compared seven datasets and found that they sometimes differ quite strongly. For example, the median volume of aerosol particles produced for coughing differs over two orders of magnitude when comparing two data sets. It is unclear what this difference represents, it might have to do with the sampling method. Furthermore, observed size distributions also differ in literature, with peak particle emission rates at different sizes. The authors should be aware that the choice of particle emission data for their study can impact their results strongly, and including discussion on the choice of data set and the implications on the results is warranted.

9. Taking 0.5 for the evaporation diameter factor is probably too large. Liu et al. (2017) find a value around one third for respiratory droplets from coughing, and the recent study on the evaporation of saliva droplets and aerosols by Lieber et al. (2021) find a value of 0.2 (for a range of temperature of 20-29 degrees C, and range of relative humidity of 6 – 65%). This likely matters for the result, as the difference between 0.5 and 0.2 leads to a factor ~15 change in droplet volume.

10. The kinetic model assumes that viral replication is controlled by the reduction of target cells over the course of the infection, but it neglects the effect of the immune system. This seems a rather strong assumption. What is the evidence for this in the case of SARS-CoV2? Also, it would be good if the authors could comment on the identifiability of the model parameters- especially the high uncertainty of the half-life of SARS-CoV2 in the respiratory tract (2.62-66hours) suggests that this might be a problem.

Additional points:

11. Line 453-454: "log 𝑘 = 𝑎(𝑆𝐷) + 𝑏, where 𝑎 is the slope for association and 𝑏 is the intercept". This appears to be a strange notation for this equation, isn't "a*SD + b" more logical?

12. Line 544-545: "To estimate the average duration of shedding, we extrapolated the model to 0 log_10_ copies/ml post-symptom onset." If the tail of the model is very long, it might take a very long time to reach 0 log_10_. Is this the case? And if yes, is perhaps a 95% decrease compared to the maximum a better measure for the duration of shedding?

13. Line 560: Is this unit correct? "𝜌 is the material density of the respiratory particle (997 g/m3)" Shouldn't this density be in kg/m3?

14. Line 563 – 571: The estimate for 𝛾 for SARS-CoV-2 could turn out to be higher, if Lednicky et al. (2020) are to be believed. In any case, this warrants some further discussion in the paper, as the results are probably quite sensitive to this parameter!

15. Line 609 – 610: "𝜌 is the material density of the respiratory particle (taken to be 1 g/cm3 based on the composition of dehydrated respiratory particles)". What is the reference for this statement? Zhang et al. (2011) find densities between 1.25 and 1.62 g/mL (Table 2). It seems logical that the drying process increases the density somewhat as compared to the density of water, as for instance the heavier salts do not evaporate.

16. Figure 2: this figure is somewhat unclear, maybe I do not understand it correctly. Why does one study have multiple standard deviations? And as there are only three values of k, regression seems to be an odd choice. A comparison between different groups of k seems more appropriate?

References:

Fears AC, Klimstra WB, Duprex P, Hartman A, Weaver SC, Plante KS, et al. 2020. Comparative dynamic aerosol efficiencies of three emergent coronaviruses and the unusual persistence of sars-cov-2 in aerosol suspensions. medRxiv:2020.2004.2013.20063784.

Lednicky JA, Lauzardo M, Fan ZH, Jutla AS, Tilly TB, Gangwar M, et al. 2020. Viable sars-cov-2 in the air of a hospital room with covid-19 patients. medRxiv:2020.2008.2003.20167395.

Lieber, C., Melekidis, S., Koch, R., Bauer, H.-J., 2021. Insights into the evaporation characteristics of saliva droplets and aerosols: Levitation experiments and numerical modeling. Journal of Aerosol Science 154, 105760.

Liu, L., Wei, J., Li, Y., Ooi, A., 2017. Evaporation and dispersion of respiratory droplets from coughing. Indoor Air 27, 179-190.

Schijven, J.F., Vermeulen, L.C., Swart, A., Meijer, A., Duizer, E., de RodaHusman, A.M., 2020. Exposure assessment for airborne transmission of SARS-CoV-2 via breathing, speaking, coughing and sneezing. medRxiv, 2020.2007.2002.20144832.

van Kampen JJA, van de Vijver DAMC, Fraaij PLA, Haagmans BL, Lamers MM, Okba N, et al. 2020. Shedding of infectious virus in hospitalized patients with coronavirus disease-2019 (covid-19): Duration and key determinants. medRxiv: 2020.2006.2008. 20125310.

Zhang, T., 2011. Study on Surface Tension and Evaporation Rate of Human Saliva, Saline, and Water Droplets. West Virginia University,

---

## [Author Response]

Essential revisions:1. The meta-regression is based on only three viral species and hence it is unclear how generalisable the observed association between rVL and transmission heterogeneity is. In the best case the data show that the three virus species exhibit significantly different rVL-variation, which coincides with their different k values at the epidemiological level. However, this latter association is essentially based on only three data points (i.e. the three viral species). The current meta-regression approach (applying a simple linear regression but essentially ignoring the fact that all studies stem from only three virus species; i.e., ignoring the hierarchical nature of the data) provides p values which strongly exaggerate the degree of evidence.

We have revised the reporting of the described analysis to address the points raised by the reviewers. We have edited the meta-regression result to specifically mention it refers to analysis of these three viruses. In addition, rather than report the specific *P*-value from our meta-regression (which was orders of magnitude lower), we now report “*P* < 0.001”, which is revised in Figure 2 as well as the text. The revised text says, “meta-regression (Figure 2) showed a strong, negative association between k and heterogeneity in rVL for these three viruses (meta-regression slope t-test: *P* < 0.001, Pearson’s *r* = -0.73).” (page 6, line 123-125). These revisions have have softened the described degree of evidence and report the results of our meta-regression as specifically based on the three viruses (SARS-CoV-2, SARS-CoV-1 and A(H1N1)pdm09).

2. One major potential confounder is the very strong dependence of rVL on infection time. The authors consider this in the section "SARS-CoV 2 kinetics during respiratory infection" where they show also a substantial variation of rVL across different strata of days from symptom onset (DFSO). However, it is unclear to what extent this affects the previous analyses (e.g., the meta-regression models). More fundamentally, the cross-sectional nature of the rVL data leads almost by necessity to an overestimation of the variability in the transmission potential of infections and in a very strong dependency of the rVL variation on the distribution of sampling time. Even a stratification on DFSO can only partly address these problems, firstly because DFSO is in most cases associated with a substantial uncertainty which in the case of a highly dynamic infection will translate into an even larger variation of rVLs. Moreover, even if DFSO were an exact measure of infection time, different infections do not need to be synchronous (e.g. because of stochastic effects or variation of the processes corresponding to the model parameters across individuals) such that different individuals will have their peak rVL at different time points. Taken together this implies that a reliable measure of heterogeneity would require determining something like the area under the curve of rVL (which of course is very challenging).

Our systematic review and data collection were designed to specifically develop a dataset of rVLs from the estimated infectious periods for each virus. While, as noted by the reviewers, the cross-sectional nature of data, like virus quantitation, means that some DFSO may be more prevalent in the dataset in each study. However, when considered together, the studies identified by the systematic review tended to span the estimated infectious period for each of the three viruses. Since the meta-analyses (before the “SARS-CoV-2 kinetics during the respiratory infection” section) are conducted cumulatively on the identified studies, they “approximated the expected rVL when encountering a COVID-19, SARS or A(H1N1)pdm09 case during the infectious period” (page 7, line 138-139). For the meta-regression, each study is an estimate of the SD of the rVLs in their analyzed period. Like described above, the meta-regression should approximate the SD of the rVLs throughout the infectious period. As a limitation of the study, we note that there were fewer studies that had data from the presymptomatic period: “The systematic search found a limited number of studies reporting quantitative specimen measurements from the presymptomatic period, meaning these estimates may be sensitive to sampling bias.” (page 15, line 399-401).

As the reviewers mentioned, there is uncertainty in DFSO. For example, DFSO is often based on patient recall, which is uncertain. Even in the scenario that DFSO is certain, then cases can have different peak rVLs on different DFSO, as the reviewers also mentioned. To discuss this potential confounder in the former point, we have written, “Specimen measurements (based on instrumentation, calibration, procedures and reagents) are not standardized and, as DFSO is typically based on patient recall, there is also inherent uncertainty in these values. While the above procedures (including only quantitative measurements after extraction as an inclusion criterion, considering assay detection limits and correcting for specimen dilution) have considered many of these factors, non-standardization remains an inherent limitation in the variability of specimen measurements.” (page 23, line 588-593). For the latter point, if rVL peaks on different DFSO for different individuals, then this variability should influence, and be shown in, our aggregate analyses and estimates. Stratification for DFSO also considers this. For example, when encountering a case on 5 DFSO, we wish to know what the expected distribution of rVLs that the case may have. Our analyses consider the distribution of rVLs on that DFSO, and thus this approach assesses rVL variability on each DFSO across cases.

3. This limitation also strongly affects the practical, public health relevance of the findings. For example, the authors state that "Our analyses suggest that heterogeneity in rVL may be generally associated with over-dispersion for viral respiratory infections. In this case, rVL distribution can serve as an early correlate for transmission patterns, including super-spreading, during outbreaks of novel respiratory viruses, providing insight for disease control before large-scale epidemiological studies empirically characterise k ". This potential application assumes that the timing of rVL measurements is known early in a pandemic and that it can be controlled for, which requires a detailed knowledge of the within patient dynamics of the virus. I would assume that achieving this knowledge would take at least as long as estimating k in epidemiological studies. Thus it may be more appropriate to think about the two approaches in characterising heterogeneity as complementary (in the context of epidemiological triangulation; i.e. both approaches having their weaknesses and biases but which can be overcome in a joint consideration; generally, I think that attempting to achieve such a triangulation is one of the main strengths of the present study, despite its limitations).

We have revised our manuscript to discuss the updated view, for which we express appreciation to the reviewers. We no longer discuss rVL as being characterized before epidemiological studies empirically characterizing k, but now write: “In this case, rVL distribution can serve as an early correlate for transmission patterns, including superspreading, during outbreaks of novel respiratory viruses. When considered jointly with contact-tracing studies, this provides epidemiological triangulation on *k*: heterogeneity in rVL indirectly estimates *k* via an association, whereas contact tracing empirically characterizes transmission chains to estimate *k* but is limited by incomplete or incorrect recall of contact events by cases.” (page 14, line 361-366).

4. The variation of rVL might also be strongly driven by the sampling method/procedure (even the same method will give very different results across health-care workers), which implies the same problems as (2) – i.e. overestimation of rVL variability and potential confounding.

The inclusion and exclusion criteria used in our systematic review protocol means that we considered a specific set of respiratory specimens from which virus quantitation was performed in a similar manner. We also accounted for variation between studies and specimen types in their processing (e.g., volume of viral transport media used) in the estimation of rVLs. Further assessment of these potential risks of bias was included through the JBI critical appraisal checklist.

We agree with the reviewers that there is increased variation in rVLs based on sampling, which is an intrinsic measurement error associated with virus quantitation. We do note this limitation with “Specimen measurements (based on instrumentation, calibration, procedures and reagents) are not standardized and, as DFSO is typically based on patient recall, there is also inherent uncertainty in these values. While the above procedures (including only quantitative measurements after extraction as an inclusion criterion, considering assay detection limits and correcting for specimen dilution) have considered many of these factors, non-standardization remains an inherent limitation in the variability of specimen measurements” (page 23, line 588-593).

Each viral load for each virus was measured in a comparable manner. Thus, the intrinsic measurement error should be similar for each virus. In other words, each virus will have a similar increase in variability from measurement error. Thus, this error should not drive the observed differences in rVL heterogeneity, and the difference in heterogeneity in rVL between viruses should arise from the viruses. We have also revised the manuscript to describe this point, “While there is intrinsic measurement error in virus quantitation, based on the systematic review protocol and study design (as described above), this error should similarly increase heterogeneity in rVL for each virus, and the difference in heterogeneity in rVL between viruses should arise from the viruses.” (page 24, line 614-617).

5. The authors note that "Talking, singing and coughing expelled virions at greater proportions via droplets (80.6-86.0%) than aerosols (14.0-19.4%)." It should be noted that although more virions are expelled via droplets than aerosols according to the findings of this study, exposure to droplets and aerosols is not equal and this could affect the probability of transmission via these routes. For example, if social distancing and masking is observed then it is possible that larger droplets are more easily captured by masks or fall on the ground quickly and do not reach a susceptible individual, while aerosols do. Furthermore, smaller droplets and aerosols can penetrate more deeply into the lungs. It is as of yet unclear whether this would influence the probability of becoming (more severely) infected. This may also differ per virus. A discussion on these issues is relevant.6. The authors also note that their results "support aerosol spread as a transmission mode for SARS-CoV-2, including for conditional superspreading by highly infectious cases. However, with short durations of stay in well-ventilated areas, the exposure risk for aerosols, including long-range and buoyant ones, remains correlated with proximity to infectious cases."

As these two comments are related, we address them together. Based on the revised evaporation diameter factor from Reviewer Comment 9, our model now estimates that “Talking, singing and coughing expelled virions at comparable proportions via droplets (55.6-59.4%) and aerosols (40.6-44.4%)” (page 11, line 276-278).

The discussion mentioned by the reviewers, however, remains relevant. We have provided a more nuanced revision on abating droplet and aerosol transmission, as exposure to both droplets and aerosols is correlated with proximity. We have revised the manuscript based on reviewer comments 5 and 6, with this nuanced perspective in mind, and it now reads: “While talking, singing and coughing, our models indicate that SARS-CoV-2 is carried by droplets (55.6-59.4% of shed virions), short-range aerosols (30.1-34.9%), long-range aerosols (7.7-8.3%) and buoyant aerosols (0.01-6.5%). Transmission, however, requires exposure. For direct transmission, droplets tend to be sprayed ballistically onto susceptible tissue, whereas aerosols can be inhaled, may penetrate more deeply into the lungs and more easily facilitate superspreading events. However, with short durations of stay in well-ventilated areas, the exposure risk for both droplets and aerosols remains correlated with proximity to infectious cases (Liu, Li, et al., 2017; Prather et al., 2020). Strategies to abate infection should limit crowd numbers and duration of stay while reinforcing distancing, low-voice amplitudes and widespread mask usage; well-ventilated settings can be recognized as lower-risk venues.” (page 16-17, line 430-453).

7. A methodological note on the modelling that may affect the results (but likely do not impact the conclusions strongly) is the following.The authors take a value of 0.1% for the fraction of SARS-CoV-2 RNA copies that represents viable virus (parameter γ). This value is quite uncertain. More literature (not yet peer-reviewed) exists on the fraction of SARS-CoV-2 RNA copies that is infectious virus, providing different values. Van Kampen et al. (2020) only found a cytopathic effect on Vero cells if the swab sample from patients contained more than 7 log_10_ RNA copies/mL. Fears et al. (2020) find an average of 0.003 (range 0.0008 – 0.02) CCID50/RNA copy. However, Lednicky et al. (2020) sampled SARS-CoV-2-containing aerosols in a room with COVID-19 patients with air samplers using a water vapor condensation mechanism and as such collect virus particles without damaging them, and found an average of 0.6 CCID50/RNA copy, much higher. The model is likely sensitive to this parameter, and this could influence the result. These issues should be taken into account in the revision.

A 𝛾 of 0.1% is “equivalent to one viable copy in 3 log_10_ copies/ml for rVL or, after dilution in transport media, roughly one in 4 log_10_ copies/ml for specimen concentration” (page 29, line 724-725). For both influenza A and SARS-CoV-2, culture-positive virus has been found in the respiratory specimens considered in our study down to 4 log_10_ copies/ml, “including from pediatric (L'Huillier et al., 2020) and asymptomatic (Arons et al., 2020) [COVID-19] cases” (page 29, line 729-730). Thus, we took 𝛾 to be “0.1% as a population-level estimate” (page 29, line 722) in our model. As the reviewer mentioned, despite the discussion above, there is still uncertainty in the estimate of 𝛾. We have revised the limitations section in our manuscript to convey this, which is quoted in the response to the next reviewer comment (#8), as we combined it with the added discussion from those reviewer comments.

As research on this topic develops and methodological advances continue to improve the characterization of 𝛾, we hope that the models introduced in our study can be used as an initial basis towards even more accurate estimations of the rate, and extent, to which respiratory activities shed infectious virus via droplets and aerosols.

8. Another methodological point is that more datasets in literature are available on the emission rates and size distribution of particles during breathing, speaking, coughing etc., than are currently used to base the model on. Schijven et al. (2020) compared seven datasets and found that they sometimes differ quite strongly. For example, the median volume of aerosol particles produced for coughing differs over two orders of magnitude when comparing two data sets. It is unclear what this difference represents, it might have to do with the sampling method. Furthermore, observed size distributions also differ in literature, with peak particle emission rates at different sizes. The authors should be aware that the choice of particle emission data for their study can impact their results strongly, and including discussion on the choice of data set and the implications on the results is warranted.

We have added discussion on this topic and in its implications (please note that we combined this with added discussion from the above reviewer comment, #7): “Furthermore, this study considered population-level estimates of the infectious periods, viability proportions and profiles for respiratory particles, which omit individual or environmental variation. […] Cumulatively, these sources of variation may influence the shedding model estimates, further increasing heterogeneity in individual infectiousness.” (page 16, line 417-429).

9. Taking 0.5 for the evaporation diameter factor is probably too large. Liu et al. (2017) find a value around one third for respiratory droplets from coughing, and the recent study on the evaporation of saliva droplets and aerosols by Lieber et al. (2021) find a value of 0.2 (for a range of temperature of 20-29 degrees C, and range of relative humidity of 6 – 65%). This likely matters for the result, as the difference between 0.5 and 0.2 leads to a factor ~15 change in droplet volume.

We have adjusted the evaporation diameter factor to “0.3”, as an approximation between the factors of 0.5 (Johnson et al., 2011), 0.32 (Liu et al., 2017) and 0.2 (Lieber et al., 2017), as described in the Methods: “In our models, we took the diameter of dehydrated respiratory particles to be 0.3 times the initial size when atomized in the respiratory tract (Johnson et al., 2011; Lieber, Melekidis, Koch, and Bauer, 2021; Liu, Wei, Li, and Ooi, 2017).” (page 31, line 767-769).

This revision led to updates in Figure 5, Figure 5—Figure supplement 1 and Figure 5—Figure supplement 2, as well as in the reporting of the model results based on particle size (see edits throughout page 10, line 207-216).

10. The kinetic model assumes that viral replication is controlled by the reduction of target cells over the course of the infection, but it neglects the effect of the immune system. This seems a rather strong assumption. What is the evidence for this in the case of SARS-CoV2? Also, it would be good if the authors could comment on the identifiability of the model parameters- especially the high uncertainty of the half-life of SARS-CoV2 in the respiratory tract (2.62-66hours) suggests that this might be a problem.

Our kinetic model (Equations. 4-6, page 26, line 671-673) is dynamical and involves one independent variable (DFSO represented by *t*); one explicit dependent variable (rVL represented by *V*); two implicit dependent variables (the number of uninfected target cells represented by *T*, and the number of productively infected cells represented by *I*); and four fitted parameters: *β* (infection rate constant)*, p* (cellular shedding rate of virus), *c* (clearance rate of virus) and *δ* (clearance rate of infected epithelial cells).

Thus, this model does account for the effect of the immune system. It accounts for viral RNA clearance by any mechanism (via *c*) or infected cells cleared by any mechanism (via *δ*), including by the immune system for both, as described by the system of equations (Equations 4-6, page 26, line 671-673).

We have revised the manuscript on SARS-CoV-2 half-life, as the reviewers mentioned, it is a confusing term when described in the body. Virus half-life in the respiratory tract was calculated by t1/2=ln2/c (page 27, line 678). As the fitted estimate of *c* was 3.30 (0.25-6.34) days^-1^, the lower range of the 95% CI is below 1. The half-life equation uses *c* in the denominator, meaning *t_1/2_* is particularly sensitive to values of *c* below 1. To reduce confusion over this, we have reported the half-life in days, as it is *c*’s original unit: “the half-life of SARS-CoV-2 RNA before clearance from the respiratory tract was 0.21 (0.11-2.75) days” (page 8, line 177-178) and both *c* and *t_1/2_* are included in Figure 4—Figure Supplement 4. We have included discussion on this: “The estimated half-life of SARS-CoV-2 RNA has a skewed 95% CI (Figure 4—Figure Supplement 4). As c is in the denominator of the equation for half-life (t1/2=ln2/c), t1/2 is sensitive to *c* below one, which is the case for its lower 95% CI (Figure 4—Figure Supplement 4) and the source of the skew.” (page 28, line 697-700).

Additional points:11. Line 453-454: "log k = a(SD) + b, where a is the slope for association and b is the intercept". This appears to be a strange notation for this equation, isn't "a*SD + b" more logical?

This equation has been modified to "a*SD + b" (page 23, line 596).

12. Line 544-545: "To estimate the average duration of shedding, we extrapolated the model to 0 log_10_ copies/ml post-symptom onset." If the tail of the model is very long, it might take a very long time to reach 0 log_10_. Is this the case? And if yes, is perhaps a 95% decrease compared to the maximum a better measure for the duration of shedding?

The tail of the model was nearly linear (Figure 4D) and should not have an extended tail for this extrapolation.

13. Line 560: Is this unit correct? "ρ is the material density of the respiratory particle (997 g/m3)" Shouldn't this density be in kg/m3?

We appreciate the reviewers for spotting this typo. The material density has been corrected to 997 kg/m^3^ (page 29, line 719).

14. Line 563 – 571: The estimate for γ for SARS-CoV-2 could turn out to be higher, if Lednicky et al. (2020) are to be believed. In any case, this warrants some further discussion in the paper, as the results are probably quite sensitive to this parameter!

We have addressed this in our response to reviewer comment 7, as both questions focus on the estimate for 𝛾. Please refer to that response.

15. Line 609 – 610: "𝜌 is the material density of the respiratory particle (taken to be 1 g/cm3 based on the composition of dehydrated respiratory particles)". What is the reference for this statement? Zhang et al. (2011) find densities between 1.25 and 1.62 g/mL (Table 2). It seems logical that the drying process increases the density somewhat as compared to the density of water, as for instance the heavier salts do not evaporate.

The respiratory particles are atomized from the extracellular fluid in the respiratory tract, we considered the particles to be hydrated during atomization. The model approximated particle material density based on the density of water while hydrated.

16. Figure 2: this figure is somewhat unclear, maybe I do not understand it correctly. Why does one study have multiple standard deviations? And as there are only three values of k, regression seems to be an odd choice. A comparison between different groups of k seems more appropriate?

Each dot in this figure represents a separate study. We took the value of *k* for each virus based on our pooled estimates of *k* from the literature. Then, we used the SD in each study as an estimate of the heterogeneity in rVL for the respective virus in estimated infectious period. Thus, we performed a meta-regression based on the SD from each study identified in our systematic review (Figure 2—Figure supplement 1) or on the SD from each study that was assessed as having low risk of bias (Figure 2).